# WingsFL: Speed-up Federated Learning via Co-optimization of Communication Frequency and Gradient Compression Ratio

## Abstract

Federated Learning (FL) relies on two key strategies to overcome communication bottlenecks, which prevent training under low bandwidths and a large number of workers. The first strategy is infrequent communication, a core feature of the FedAVG algorithm, controlled by the number of local steps $\tau$. The second is gradient compression, a widely-used technique to reduce data volume, governed by a compression ratio $\delta$. However, finding the optimal $(\tau, \delta)$ pair is a major challenge in realistic settings with device heterogeneity and network fluctuations. Existing work assumes that the effects of $\delta$ and $\tau$ on the model convergence are orthogonal, optimizing them separately. In this work, we challenge this orthogonality assumption. We are the first to propose two virtual queues at distinct temporal granularities, helping derive the bounds of the noise introduced by the two lossy strategies, respectively. We demonstrate that the convergence rate of FedAVG with gradient compression is critically affected by a key term $2^\tau/\delta^2$. This finding proves that $\tau$ and $\delta$ are intrinsically coupled and must be co-designed for efficient training. Furthermore, we propose WingsFL, which fixes the key convergence rate term and minimizes the end-to-end training time under device heterogeneity by solving a one-variable Min-Max problem. WingsFL achieves up to $2.24\times$ and $2.18\times$ speed-ups over FedAVG and SOTA adaptive strategies, respectively, considering device heterogeneity and network fluctuations.

## 1 Introduction

Federated Learning (FL) has become a popular framework in modern Distributed Machine Learning (DML), but there are communication bottlenecks during the training process. FL enables collaborative model training across distributed nodes while preserving data privacy Ye et al. (2023); Dong et al. (2025). As the model parameters grow explosively, particularly large-scale models Achiam et al. (2023), the volume of gradients exchanged during FL training has become a significant communication bottleneck Lu et al. (2025b). The bottlenecks severely prevent the training process when encountering low bandwidth or a larger number of nodes Lin et al. (2018). To mitigate this, infrequent communication and gradient compression are employed. Infrequent communication is intrinsic to the classic FedAVG algorithm McMahan et al. (2017), which reduces the total number of communication rounds by performing $\tau$ local training steps before each aggregation. Sparsification gradient compression Wang et al. (2023); Lu et al. (2024) is widely adopted to reduce the communication volume of each individual communication round through transferring gradient elements with compression ratio $\delta$ Lu et al. (2025b). We denote FedAVG with sparsification gradient compression as FedAVG-GC, involving: 1) workers perform $\tau$ local iterations to compute updates; 2) workers compress the updates by a ratio $\delta$ and upload them; 3) the server aggregates the compressed updates in the same way as FedAVG to have the updated global model and broadcast the new model to workers.

The compression ratio $\delta$ and local iterations $\tau$ are critical communication hyperparameters, but the selection of them is difficult. The selection of $\delta$ and $\tau$ both face a critical trade-off. The overly aggressive strategy (large $\tau$ or small $\delta$) will cause accuracy degradation, especially in non-IID scenarios, while the conservative strategy will face expensive communication cost Hsieh et al. (2020); Li et al. (2022). Also, $\delta$ tends to adaptively change with the dynamic bandwidth, can speed up to $4.1\times$ over the static strategy Abdelmoniem & Canini (2021b). The selection of $\tau$ should take into account

the heterogeneity of the device, to avoid a severe straggler effect Li et al. (2020). The selection of $\delta$ and $\tau$ is somewhat similar (although not totally the same), while previous works neglect this similarity, assuming the gradient compression is orthogonal to the infrequent communication Cui et al. (2021); Liu et al. (2022). We have two problems: 1) Is the effect of $\delta$ orthogonal to $\tau$ on the convergence rate of FedAVG-GC? 2) If not, can we use the nature to speed up FedAVG-GC training end-to-end?

In response to the first question, our theoretical analysis shows that they are not orthogonal. Inspired by the Nested Virtual Sequence (NVS) proposed in previous work Lu et al. (2025a), we propose a framework called NVS-FL, which introduces two virtual sequences at two distinct scales: local computation and global aggregation, resolving the issue that NVS could only introduce multiple virtual sequences within the same time scale according to its framework. The virtual sequence from the local computation decouples the error bound introduced by infrequent communication, while the other one decouples the impact of gradient compression. Based on this, we derive the convergence rate of FedAVG-GC, where our analysis is the first to identify a critical term $\frac{2^\tau}{\delta^2}$, which governs the number of local iterations required to reach a target accuracy. This result reveals a theoretical equivalence between the impact of local iterations, through the term $2^\tau$, and gradient compression, through the term $\delta^{-2}$ on model convergence.

To address the second question, we establish a mathematical model for the end-to-end training time that accounts for device heterogeneity and dynamic network bandwidth. Using this model, we formulate the task of minimizing training time as a dual-variable Min-Max problem. Leveraging our theoretical equivalence, this problem is simplified to finding the minimum of a one-variable piecewise function. We propose WingsFL, which uses a standard binary search algorithm per several iterations to choose an optimal $(\delta, \tau)$ pair in challenging dynamic network conditions and device heterogeneity. In WingsFL, we accelerate FL training end-to-end by co-optimizing $\delta$ and $\tau$, like two wings of the bird.

Our main contributions are as follows:

- We propose NVS-FL, a novel theoretical framework for analyzing FedAVG-GC, by setting two virtual sequences under different timescales. Our analysis is the first to theoretically establish that the gradient compression and infrequent communication are not orthogonal strategies in FedAVG. The results theoretically reveal that $\frac{2^\tau}{\delta^2}$ governs the convergence bound in non-IID scenarios.

- We mathematically model the end-to-end training time under device heterogeneity and dynamic bandwidth. Building on this model and our theoretical insights, we propose WingsFL, which jointly optimizes $\delta$ and $\tau$ to accelerate FL training.

- We conduct extensive experiments under device heterogeneity (different computing speeds and network conditions) and dynamic network environments across diverse model architectures, including CNN, VGG, GPT, and ViT. Our results demonstrate that WingsFL achieves up to $2.24\times$ and $2.18\times$ speed-ups over static and SOTA adaptive strategies, respectively.

## 2 Preliminaries

### 2.1 Federated Learning

The primary goal of FL is to collaboratively train a global model by minimizing a global objective function $f(\mathbf{x})$, which is typically defined as a weighted average of the local objective functions $f_i(\mathbf{x})$ from $n$ participating workers:

$$\min_{\mathbf{x} \in \mathbb{R}^d} f(\mathbf{x}) = \sum_{i=1}^{n} p_i f_i(\mathbf{x}),$$

where $\mathbf{x} \in \mathbb{R}^d$ represents the model parameters, $d$ is the model dimension, and $p_i$ is the weight of the $i$-th worker, usually proportional to its local dataset volume, such that $p_i > 0$ and $\sum_{i=1}^{n} p_i = 1$. Each local objective function $f_i(\mathbf{x})$ is the expected loss over the worker's local data distribution.

### 2.2 Challenges in Real-World Federated Learning

Non-IID datasets, device heterogeneity, and constrained network scenarios are three challenges in real-world FL training.

- *Non-IID datasets:* In FL, worker data is typically not independent and identically distributed (Non-IID). This introduces a discrepancy between the local worker objectives and the global objective, which can slow down or even prevent model convergence.

- *Device heterogeneity:* workers in an FL network often possess vastly different computational capabilities (*e.g.*, CPU/GPU types, power state). This means that the time required to perform the same computational task varies significantly across devices. We model this by defining $t_{\text{latency}}^i$ as the wall-clock time for worker $i$ to complete a single local training step (i.e., one forward and backward pass). A system with a high variance in $\{t_{\text{latency}}^i\}_{i=1}^n$ is considered highly heterogeneous from a computational standpoint.

- *Constrained network scenarios:* In this scenario, device connectivity is often unstable and heterogeneous. Factors such as the connection type (Wi-Fi vs. cellular), network congestion, and physical distance to the server cause communication speeds to fluctuate widely. We assume that each worker has a different and time-varying upload bandwidth. In contrast to stable, high-speed datacenter interconnects, these network environments are characterized by high-latency and low-bandwidth. Training across WAN is a typical case, which has an average bandwidth often below 1Gbs.

## 2.3 Communication Optimization Methods in FL

### 2.3.1 Infrequent Communication

The FedAVG algorithm McMahan et al. (2017) introduces the concept of infrequent communication, where workers perform $\tau > 1$ local steps of SGD before communicating with the server. This means that each worker computes $\tau$ forward and backward passes every global iteration. This reduces the total number of communication rounds by $\tau$ times during the same local iterations.

### 2.3.2 Gradient Compression

Gradient compression techniques Xu et al. (2021) are generally grouped into three main types: (1) sparsification, which involves sending only a subset of gradient entries; (2) quantization, where high-precision values are transformed into lower-precision representations; and (3) low-rank approximation, which expresses the gradient as the product of two low-rank matrices. Among these, sparsification is often favored for its effectiveness in eliminating redundant gradient components and achieving higher communication efficiency. Additionally, compared to other compressors, sparsification compressors offer a continuous range of compression ratios, facilitating adaptive optimization. Sparsification compressors include the relative compressor, like Top-$k$ and Random-$k$ Abdelmoniem & Canini (2021b), and the absolute compressor, like the hard-threshold compressor Sahu et al. (2021). Due to that, the absolute compressor does not perform well in FL Lu et al. (2025b), so we don't take it into consideration. The sprasification compression usually comes with Error-Feedback (EF) Dorfman et al. (2023); Stich & Karimireddy (2020a), a popular mechanism that collects and reuses the errors from the gradient compression to mitigate the compression bias and guarantee convergence. For the update algorithm, at the global iteration $T$, each worker $i$ maintains an error term $\mathbf{e}_T^i$ and has the update $\Delta_T^i$ (accumulated in $\tau$ local iterations), then the worker $i$ gets the compressed update $\hat{\Delta}_T^i = \mathbf{C}_\delta(\Delta_T^i + \mathbf{e}_T^i)$, which will be sent to the server, and updates its error term $\mathbf{e}_{T+1}^i = \mathbf{e}_T^i + \Delta_T^i - \hat{\Delta}_T^i$.

## 3 Theoretical Analysis of FedAVG-GC

We derive the convergence rate of FedAVG-GC and list the analysis in remarks below the theorem. Notation list and detailed proof are in the Appendix. The pseudocode of FedAVG-GC is shown in Algo. 3 in the Appendix.

### 3.1 Regular Assumptions

**Theorem 2** and **Theorem 3** is established under the assumption that the objective functions are $\mu$-strongly convex. A detailed list of assumptions is outlined below.

**Assumption 1** (*L*-smoothness). We assume *L*-smoothness of $f_i$, $i \in [n]$, that is for all $\mathbf{x}$, $\mathbf{y} \in \mathbb{R}^d$:

$$\|\nabla f_i(\mathbf{y}) - \nabla f_i(\mathbf{x})\| \leq L\|\mathbf{y} - \mathbf{x}\|. \tag{1}$$

**Assumption 2** (Bounded gradient noise). We assume the availability of stochastic gradient oracles $\mathbf{g}_t^i : \mathbb{R}^d \to \mathbb{R}^d$ corresponding to each local objective $f_i$ for $i \in [n]$. Let $\xi^i$ denote the stochastic gradient noise introduced by the $i$-th worker. For simplicity, we focus on the representative scenario where $\xi^i$ is uniformly bounded across all $\mathbf{x} \in \mathbb{R}^d$ and all clients $i \in [n]$:

$$\mathbf{g}_t^i = \nabla f_i(\mathbf{x}_t) + \boldsymbol{\xi}^i, \qquad \mathbb{E}_{\boldsymbol{\xi}^i}\boldsymbol{\xi}^i = \mathbf{0}_d, \qquad \mathbb{E}_{\boldsymbol{\xi}^i}\|\boldsymbol{\xi}^i\|^2 \leq \sigma^2. \tag{2}$$

**Assumption 3** (Measurement of data heterogeneity). We quantify the level of data heterogeneity using a non-negative constant $\zeta^2 \geq 0$, which serves as an upper bound on the variance among the $n$ nodes. Specifically, we assume:

$$n \sum_{i \in [n]} p_i^2 \|\nabla f_i(\mathbf{x})\|^2 \leq \zeta^2 + Z^2 \|\nabla f(\mathbf{x})\|^2, \quad \forall \mathbf{x} \in \mathbb{R}^d, i \in [n]. \tag{3}$$

**Assumption 4** ($\mu$-strongly convexity). We assume $\mu$-strong convexity of $f_i, i \in [n]$, that is for all $\mathbf{x}, \mathbf{y} \in \mathbb{R}^d$:

$$f_i(\mathbf{x}) - f_i(\mathbf{y}) \geq \langle \nabla f_i(\mathbf{y}), \mathbf{x} - \mathbf{y} \rangle + \frac{\mu}{2}\|\nabla f_i(\mathbf{x}) - \nabla f_i(\mathbf{y})\|^2. \tag{4}$$

## 3.2 Theoretical Framework in FL

NVS framework Lu et al. (2025a) is proposed to convert the convergence of complex SGD variants into a standard SGD process and several analyzable noise terms, which can be derived from the bound of the term. However, it assumes homogeneous iteration intervals, which is unsuitable for FL scenarios. In FL, the update process is typically heterogeneous: local models undergo frequent updates (*e.g.*, every iteration), while the global model aggregates these updates infrequently (*e.g.*, every $\tau$ iterations). To address this challenge, we introduce a novel theoretical framework for FL called NVS-FL, which defines two distinct virtual sequences to handle the different timescales of local training and global aggregation.

In detail, we define $\mathbf{x}_0 = \tilde{\mathbf{x}}_0 = \hat{\mathbf{x}}_0$ and define the first virtual sequence in the global aggregation timescale:

$$\mathbf{x}_{T+1} = \mathbf{x}_T - \mathbf{v}_T, \quad B_{T+1} = B_T + \gamma \sum_{i \in [n]} p_i \Delta_T - \mathbf{v}_T, \quad \tilde{\mathbf{x}}_T = \mathbf{x}_T - B_T. \tag{5}$$

In this way, we can derive that $\tilde{\mathbf{x}}_{T+1} = \mathbf{x}_{T+1} - B_{T+1} = \mathbf{x}_T - \mathbf{v}_T - B_T - \gamma \sum_{i \in [n]} p_i \Delta_T^i + \mathbf{v}_T = \tilde{\mathbf{x}}_T - \gamma \sum_{i \in [n]} p_i \Delta_T^i$. Then we define the second virtual sequence in the local iteration timescale. We define $\hat{\mathbf{x}}_{T,\tau} = \hat{\mathbf{x}}_{T+1,0} = \tilde{\mathbf{x}}_{T+1}$ and have:

$$\hat{\mathbf{x}}_{T,j+1} = \hat{\mathbf{x}}_{T,j} - \gamma \sum_{i \in [n]} p_i \mathbf{g}_{T,j}^i. \tag{6}$$

In FedAVG-GC, $\mathbf{v}_T = \gamma \sum_{i \in [n]} p_i \tilde{\Delta}_T^i$, so we have $B_T = \gamma \sum_{i \in [n]} p_i \mathbf{e}_T^i$.

## 3.3 Convergence Rate of FedAVG-GC

**Theorem 1** (Non-convex convergence rate of FedAVG-GC). *Let $f : \mathbb{R}^d \to \mathbb{R}$ be $L$-smooth. There exists a stepsize $\gamma \leq \min\{\frac{1}{2\sqrt{2}\tau L}, \frac{1}{\sqrt{32\phi}\tau LZ}\}$, where $\phi = \frac{2^\tau}{\delta^2}$, such that at most*

$$\mathcal{O}\left(\frac{\sum_{i \in [n]} p_i^2 \sigma^2}{\epsilon^2} + \frac{\sqrt{\phi\zeta^2 + (1 + n\phi\delta)\sum_{i \in [n]} p_i^2 \sigma^2}}{\epsilon^{3/2}} + \frac{1}{\epsilon} + \frac{Z\sqrt{\phi}}{\epsilon}\right) \cdot L(f(\mathbf{x}_0) - f^*) \tag{7}$$

*local iterations of FedAVG-GC, it holds $\mathbb{E}\|\nabla f(\mathbf{x}_{out})\|^2 \leq \epsilon$, and $\mathbf{x}_{out} = \mathbf{x}_t$ denotes an iterate $\mathbf{x}_t \in \{\mathbf{x}_0, \ldots, \mathbf{x}_{(\tau \cdot T_{end}-1)}\}$, where $\tau \cdot T_{end}$ denotes the total number of local iterations, chosen at random uniformly.*

**Theorem 2** (Convex convergence rate of FedAVG-GC, *i.e.*, $\mu = 0$). *Let $f : \mathbb{R}^d \to \mathbb{R}$ be $L$-smooth and $\mu$-convex. Then there exists a stepsize $\gamma \leq \min\{\frac{1}{4L}, \frac{1}{4\sqrt{3}L\tau}, \frac{1}{\sqrt{384 \cdot \phi}\tau LZ}\}$, where $\phi = \frac{2^\tau}{\delta^2}$, such that at most*

$$\mathcal{O}\left(\frac{\sum_{i \in [n]} p_i^2 \sigma^2}{\epsilon^2} + \frac{\sqrt{L\phi\zeta^2 + L(1 + n\phi\delta)\sum_{i \in [n]} p_i^2 \sigma^2}}{\epsilon^{3/2}} + \frac{L}{\epsilon} + \frac{ZL\sqrt{\phi}}{\epsilon}\right) \cdot \|\mathbf{x}_0 - \mathbf{x}_*\|^2 \tag{8}$$

*local iterations of FedAVG-GC, it holds $\mathbb{E}f(\mathbf{x}_{out}) - f^* \leq \epsilon$, and $\mathbf{x}_{out} = \mathbf{x}_t$ denotes an iterate $\mathbf{x}_t \in \{\mathbf{x}_0, \ldots, \mathbf{x}_{(\tau \cdot T_{end}-1)}\}$, where $\tau \cdot T_{end}$ denotes the total number of local iterations, chosen at random uniformly.*

**Theorem 3** ($\mu$-strongly convex convergence rate of FedAVG-GC, *i.e.*, $\mu > 0$). *Let $f : \mathbb{R}^d \to \mathbb{R}$ be $L$-smooth and $\mu$-convex. Then there exists a stepsize $\gamma \leq \min\{\frac{1}{4L}, \frac{1}{4\sqrt{3}L\tau}, \frac{1}{\sqrt{384 \cdot \phi \tau LZ}}\}$, where $\phi = \frac{2^\tau}{\delta^2}$, such that at most*

$$\mathcal{O}\left(\frac{\sum_{i \in [n]} p_i^2 \sigma^2}{\mu\epsilon} + \frac{\sqrt{L\phi\zeta^2 + L(1+n\phi\delta)\sum_{i \in [n]} p_i^2 \sigma^2}}{\mu\epsilon^{1/2}} + \frac{L}{\mu} + \frac{ZL\sqrt{\phi}}{\mu}\right) \cdot \|\mathbf{x}_0 - \mathbf{x}_*\|^2 \quad (9)$$

*local iterations of FedAVG-GC, it holds $\mathbb{E}f(\mathbf{x}_{out}) - f^* \leq \epsilon$, and $\mathbf{x}_{out} = \mathbf{x}_t$ denotes an iterate $\mathbf{x}_t \in \{\mathbf{x}_0, \ldots, \mathbf{x}_{(\tau \cdot T_{end}-1)}\}$, where $\tau \cdot T_{end}$ denotes the total number of local iterations, selected probabilistically based on $(1 - \min\{\frac{\mu\gamma}{2}, \frac{\delta}{2 \cdot 2^\tau}\})^{-t}$.*

### 3.4 ANALYSIS OF CONVERGENCE RATE IN FL

Given that the convergence rate exhibits similar expressions under convex and non-convex cases, we refer to prior workLu et al. (2024) and take **Theorem 1** as an example for analysis.

**Remark 1.** ($\phi$ determining the convergence in non-IID scenarios) Due to the The fisrt term $\frac{\sum_{i \in [n]} p_i^2 \sigma^2}{\epsilon^2}$ is determined by the task, independent to $\delta$ and $\tau$. The magnitude of the second term $\frac{\sqrt{\phi\zeta^2 + (1+\phi\delta)\sum_{i \in [n]} p_i^2 \sigma^2}}{\epsilon^{3/2}}$ is greater than that of the third term, since $\epsilon$ is often set less than $10^{-4}$. Then we focus on the second term. FL is a typical non-IID sceanrio, where $\zeta$ is usually larger than the gradient noise $\sigma$, and the convergence rate can be written as $\mathcal{O}(\frac{\sum_{i \in [n]} p_i^2 \sigma^2}{\epsilon^2} + \frac{\sqrt{\phi\zeta^2}}{\epsilon^{3/2}})$, similar to the previous work Lu et al. (2023b). In this case, $\phi = \frac{2^\tau}{\delta^2}$ determines the convergence rate bound. For the analysis in IID scenarios, such as LLM pre-training inside a datacenter Xu et al. (2021), $\zeta^2 \ll \sigma^2$ and the convergence rate is bounded by $\phi' = \frac{2^\tau}{\delta}$. In this case, we can use $\phi'$ to design the following algorithm, but we do not discuss this in this paper.

**Remark 2.** (Degradation condition) When $\tau = 1$, FedAVG-GC degrades to Distributed SGD with gradient compression. In this case, $\phi = \frac{2}{\delta^2}$ and the convergence rate is equal to $\mathcal{O}(\frac{\sum_{i \in [n]} p_i^2 \sigma^2}{\epsilon^2} + \frac{\sqrt{2\zeta^2/\delta^2 + (1+2/\delta)\sum_{i \in [n]} p_i^2 \sigma^2}}{\epsilon^{3/2}} + \frac{\tau LZ}{\delta\epsilon})$, the same as the convergence rate of the previous work Stich (2020); Lu et al. (2023b).

## 4 WINGSFL: JOINT OPTIMIZATION OF LOCAL STEPS AND COMPRESSION

For worker $i$, we define the time for a single computation pass as $t^i_{\text{compute}}$, including one forward pass and backward pass. The end-to-end communication latency for long-distance transmission is represented as $t^i_{\text{latency}}$. We introduce a coefficient $a^i$ to characterize the time associated with gradient compression, and $b^i_T$ represents the network bandwidth at the global communication iteration $T$. The notation list is shown in the Appendix.

In each global communication round, which occurs after $\tau$ local iterations, the total computation time is given by $\tau t^i_{\text{compute}}$. The communication time for each worker $i$ in this round is formulated as $t^i_{\text{latency}} + q(\delta) + S_g\delta/b^i_T$, where $S_g$ is the size of the gradient (bits) and $q(\delta)$ is the compression cost, related to the property of the compressor. In this way, the average time for local computation and communication per iteration $T_{\text{avg}} = \frac{\tau t^i_{\text{compute}} + (t^i_{\text{latency}} + q(\delta) + S_g\delta/b^i_T)}{\tau}$. Our primary objective is to minimize end-to-end time for the entire training process. Considering the device heterogeneity, our objective can be formulated as $\min_{\tau \in [R_\tau], \delta \in (0,1]} \max_{i \in [n]} \frac{\tau t^i_{\text{compute}} + (t^i_{\text{latency}} + q(\delta) + S_g\delta/b_i)}{\tau}$. $\phi = \frac{2^\tau}{\delta^2}$ governing the convergence rate, we fix this term as one hyper-parameter $\phi_c$ and have $\tau = 2\log_2 \delta + \log_2 \phi$. With this, we can write the objective problem from a dual-variable Min-Max problem into a one-variable Min-Max problem at the given iteration $T$, as $\min_{\tau \in [R_\tau], \delta \in (0,1]} \max_{i \in [n]} Q^i_T(\tau)$.

In this work, we use the relative compressor Top-$k$, where $q(\delta) = a^i \log_2(\delta)$, and we have

$$Q_i(\tau) = t_{\text{compute}}^i + \frac{a^i}{2} + \frac{t_{\text{latency}}^i - \frac{a^i}{2}\log_2 \phi_c}{\tau} + \frac{S_g}{b_T^i \sqrt{\phi_c}} \cdot \frac{2^{\tau/2}}{\tau}. \tag{10}$$

---

**Algorithm 1: WingsFL**

**Input:** $n, [p_i], [\gamma_T], T_{\text{end}}, S_g, R_\tau,$
$\quad\quad [t_{\text{compute}}^i, t_{\text{latency}}^i, a^i, b_0^i], \phi_c$, search
$\quad\quad$ frequency $E$, the compressor $C(\cdot)$
**Output:** $\mathbf{x}_{T_{\text{end}}}$
Initialize $\mathbf{x}_0, \mathbf{e}_0^i = \mathbf{0}_d$;
**for** $T \in [T_{end}]$ **do**
$\quad$ /\* Update $\delta$ and $\tau$ per $E$
$\quad\quad$ global iterations $\quad\quad$ \*/
$\quad$ **if** $T \bmod E == 1$ **then**
$\quad\quad$ $\delta, \tau = \text{SearchAlgo}(S_g, R_\tau, \phi_c,$
$\quad\quad\quad n, [t_{\text{compute}}^i, t_{\text{latency}}^i, a^i, b_{T-1}^i]);$
$\quad$ **end**
$\quad$ /\* Worker side $\quad\quad\quad$ \*/
$\quad$ **for** $i \in [n]$ **do**
$\quad\quad$ Monitor the network condition and
$\quad\quad\quad$ update the bandwidth $b_T^i$;
$\quad\quad$ $\mathbf{x}_{T,0}^i = \mathbf{x}_T$;
$\quad\quad$ **for** $j \in [\tau]$ **do**
$\quad\quad\quad$ $\mathbf{x}_{T,j+1}^i = \mathbf{x}_{T,j}^i - \gamma_T \mathbf{g}_{T,j}^i$;
$\quad\quad$ **end**
$\quad\quad$ $\Delta_T^i = \sum_{j\in[\tau]} \mathbf{g}_{T,j}^i$;
$\quad\quad$ $\hat{\Delta}_T^i = \mathbf{C}_\delta(\Delta_T^i + \mathbf{e}_T^i)$;
$\quad\quad$ $\mathbf{e}_{T+1}^i = \mathbf{e}_T^i + \Delta_T^i - \hat{\Delta}_T^i$;
$\quad\quad$ Upload $\hat{\Delta}_T^i$ to the server;
$\quad$ **end**
$\quad$ /\* Server side $\quad\quad\quad\quad$ \*/
$\quad$ $\mathbf{x}_{T+1} = \mathbf{x}_T - \gamma_T \sum_{i\in[n]} p_i \hat{\Delta}_T^i$;
$\quad$ Broadcast $\mathbf{x}_{T+1}$;
**end**
**Return** $\mathbf{x}_{T_{\text{end}}}$;

---

**Algorithm 2: SearchAlgo**

**Input:** $S_g, R_\tau, \phi_c, n,$
$\quad\quad [t_{\text{compute}}^i, t_{\text{latency}}^i, a^i, b_{T-1}^i]$
**Output:** $\tau_T, \delta_T$
left = 1, right = $R_\tau$;
**while** *left < right-1* **do**
$\quad$ $\tau_{\text{mid}} = \lfloor \frac{\text{left+right}}{2} \rfloor$;
$\quad$ $Q_{\max} = \max\limits_{i\in[n]} Q_i(\tau_{\text{mid}})$, $Q_i$ is computed
$\quad\quad$ based on Eq. 10 if using Top-$k$;
$\quad$ Record $i_{\max}$ as the index of $Q_{\max}$;
$\quad$ /\* Determining the sign of
$\quad\quad$ the derivative of
$\quad\quad$ $Q_{i_{\text{max}}}(\tau_{\text{mid}})$ $\quad\quad\quad$ \*/
$\quad$ **if** $Q_{i_{max}}'(\tau_{mid}) < 0$ **then**
$\quad\quad$ left = $\tau_{\text{mid}}$ ;
$\quad$ **end**
$\quad$ **else if** $Q_{i_{max}}'(\tau_{mid}) > 0$ **then**
$\quad\quad$ right = $\tau_{\text{mid}}$;
$\quad$ **end**
$\quad$ **else**
$\quad\quad$ **Return** $\tau_{\text{mid}}, \min\{1, \frac{2^{\tau_{\text{mid}}/2}}{\sqrt{\phi_c}}\}$;
$\quad$ **end**
**end**
**if** $\max\limits_{i\in[n]} Q_i(\textit{left}) < \max\limits_{i\in[n]} Q_i(\textit{right})$ **then**
$\quad$ $\tau_T$ = left;
**end**
**else**
$\quad$ $\tau_T$ = right;
**end**
**Return** $\tau_T, \delta_T = \min\{1, \frac{2^{\tau_T/2}}{\sqrt{\phi_c}}\}$;

---

$Q_T^i(\cdot)$ can be written in the format: $Q_T^i(\tau) = c_1^i + \frac{c_2^i + c_{3,T}^i \cdot 2^{\tau/2}}{\tau}$. For each worker $i$, during the whole training, $c_1^i$ and $c_2^i$ are fixed under given $\phi_c$, and $c_{3,T}^i$ is correlated with the dynamic bandwidth and we have the theorem.

**Theorem 4** (Global Minimum of Maximum Convex Functions). *Let $Q_T^1(\tau), Q_T^2(\tau), \ldots, Q_T^n(\tau)$ has the form: $Q_i(\tau) = c_1^i + \frac{c_2^i + c_{3,T}^i \cdot 2^{\tau/2}}{\tau}$, with constants $c_1^i, c_2^i, c_{3,T}^i > 0$ and $\tau \geq 1$. We assume that $\tau$ is continuous and define $Q(\tau) = \max\limits_{i\in[n]} Q_T^i(\tau)$. Then $Q(\tau)$ is a convex function and any local minimum of $Q(\tau)$ is a global minimum.*

Based on **Theorem 4**, we can use a binary search algorithm to find the local minimal value of $Q(\tau) = \max\limits_{i\in[n]} Q_T^i(\tau)$, getting optimal $\delta_T, \tau_T$ for each global communication iteration $T$, shown in Algo. 2. This algorithm searches within the given interval $[1, R_\tau]$. At each global communication iteration, it evaluates the new compression ratio and communication infrequency if the bandwidth changes significantly, with the time complexity of $O(n \log R_\tau)$.

Table 1: Summary of experimental settings.

| Task | Model | # Parameters | Dataset | Non-IID Setting | $\gamma_0$ | Batch Size | $n$ | Metric | Iteration |
|------|-------|--------------|---------|-----------------|------------|------------|-----|--------|-----------|
| CV | CNN | 235,690 | CIFAR-10 | $\#C = 3$ | $4 \times 10^{-3}$ | 8 | 10 | | 20,000 |
| | VGG-11 | 9,750,922 | Flickr | Real-world | $4 \times 10^{-3}$ | 8 | 10 | Accuracy | 20,000 |
| | ViT | 86,000,000 | ImageNet | $\#C = 300$ | $4 \times 10^{-5}$ | 4 | 5 | | 1,000 |
| NLP | GPT-2 | 124,000,000 | Wikitext | Dirichlet (0.5) | $5 \times 10^{-2}$ | 3 | 5 | Perplexity | 2,000 |

Note: $n$ denotes the number of workers; $\gamma_0$ is the initial learning rate.

## 5 EVALUATION EXPERIMENTS

### 5.1 EXPERIMENTAL ENVIRONMENT

All experiments are performed on a server running Ubuntu 24.04 LTS, equipped with an Intel Xeon Gold 6230 processor and 8 Nvidia RTX 3090 GPUs, each featuring 24GB of VRAM. The software environment is built upon Python 3.10.16, with all dependencies aligned accordingly. We adopt PyTorch 2.5.1 as the core deep learning framework, utilizing CUDA 12.4 for GPU acceleration. Communication across devices is facilitated using the Gloo backend.

### 5.2 EXPERIMENTAL SETTINGS

This part shows the basic experimental setting in this section. Detailed settings and hyper-parameters are provided in the Appendix.

We validate our method across a range of computer vision and natural language processing tasks using representative models such as CNN LeCun et al. (1998), VGG11 Simonyan & Zisserman (2014), ViT Dosovitskiy (2020), and GPT Radford et al. (2019). The evaluation spans datasets including Flickr Hsieh et al. (2020), CIFAR-10 Krizhevsky et al. (2009), ImageNet Deng et al. (2009), and Wikitext Merity et al. (2022). The experimental configurations are detailed in Table 1.

**Baselines:** Top-$k$ is adopted as the standard compression technique. For the WingsFL framework with integrated Top-$k$, we benchmark against several baseline approaches: FedAVG-GC, $\gamma$-FedHT Lu et al. (2025b), PASGD, and the standard FedAVG algorithm. FedAVG-GC refers to the setting where FedAVG incorporates Top-$k$ with a fixed compression rate $\delta$. $\gamma$-FedHT represents the current SOTA adaptive gradient sparsification method in federated learning. PASGD serves as the leading adaptive communication frequency algorithm; we compare it with WingsFL under a consistent Top-$k$ compression with fixed $\delta$. All baseline methods are initialized with the same hyperparameters for compression rate and communication interval, aligning with the initial values $\delta_1$ and $\tau_1$ employed by WingsFL.

**Device Heterogeneity and non-IID settings:** To simulate device heterogeneity, we assign different computation times $t^i_{\text{compute}}$ to each client. Let $t$ denote the baseline end-to-end computation time. Under a heterogeneity level of $q\%$, the computation time for each node is uniformly sampled from the interval $[t, (1 + q\%)t]$. For instance, with 10 nodes and a heterogeneity level of $100\%$ ($q = 100$), the computation times are selected as 10 equidistant values within $[t, 2t]$. The default heterogeneity level is set to $100\%$ for all tasks. For the non-IID data partitioning, we adopt the following configurations: for VGG11@Flickr, we use the real-world non-IID settings, which is the inherent properties of the dataset itself. For other tasks, we use artificially non-IID settings, refering to previous work Li et al. (2022). The detailed non-IID settings is shown in Table 1.

### 5.3 COMPARISON OF TRAINING SPEED

As illustrated in Fig. 1, WingsFL demonstrates significant improvements in training speed compared to FedAVG-GC, $\gamma$-FedHT, PASGD, and vanilla FedAVG across all tasks. Taking VGG@Flickr as an example, when the accuracy reaches $60\%$, WingsFL achieves a $1.44\times$ speedup over FedAVG, $1.47\times$ speedup over FedAVG-GC, and $1.16\times$ speedup over $\gamma$-FedHT. At $65\%$ accuracy, the speedup values change to $1.43\times$, $1.36\times$, and $1.06\times$, respectively. The performance advantage of WingsFL stems from its fundamental design principles. Unlike existing approaches that do not explicitly optimize for end-to-end training speed, WingsFL is specifically designed to minimize the end-to-end training

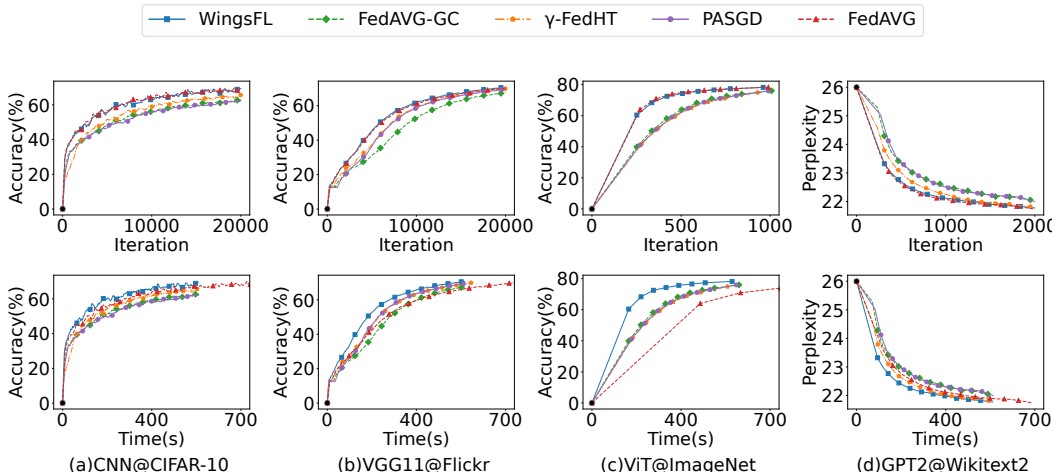

Figure 1: Training curves (Top: Accuracy vs. Iteration; Bottom: Accuracy vs. Time) across different tasks (from left to right). In all cases, WingsFL outperforms other baselines on end-to-end training.

time. Previous methods typically optimize compression ratio ($\delta$) and communication frequency ($\tau$) separately, which leads to suboptimal solutions. In contrast, our framework employs joint optimization of these parameters, guided by theoretical analysis. This coordinated approach ensures convergence while prioritizing end-to-end training efficiency, making WingsFL particularly suitable for scenarios with device heterogeneity, non-IID datasets, and dynamic network conditions.

### 5.4 SENSITIVITY ANALYSIS OF HYPERPARAMETER IN WINGSFL

Unlike most methods that heavily depend on carefully tuned hyperparameters, WingsFL exhibits low sensitivity to its key hyperparameter $\phi_c$. In this section, we examine the impact of different $\phi$ values on the end-to-end training time. The corresponding results are reported in Table 2, and we find that our design outperforms all other algorithms under the given $\phi_c$ settings. WingsFL can achieve speedups up to $1.42\times$ over FedAVG (at $\phi(30, 1\%)$, accuracy 55% in VGG@Flickr), $1.99\times$ over $\gamma$-FedHT (at $\phi(30, 0.1\%)$, ppl 22.5 in GPT@Wiki), $1.92\times$ over PASGD (at $\phi(30, 1\%)$, ppl 23.5 in GPT@Wiki).

### 5.5 WINGSFL UNDER DIFFERENT DEVICE HETEROGENEITY

The performance of algorithms under different heterogeneity levels $q$ is shown in Table 3, where the training time of baselines to converge to the target matrics is shown using the previous settings except heterogeneity levels. Our design performs better than other algorithms under different heterogeneity levels. In detail, WingsFL nearly achieves less speedups when facing higher heterogeneity levels (from $2.24\times$ to $1.38\times$ compared to FedAVG in GPT@Wiki). This is due to the fact that the higher heterogenity will not change $(\delta, \tau)$ selected from WingsFL, and the larger computation time will enlarge the partition ratio of computation time, and the gain of our design is from the reduction of communication time.

## 6 RELATED WORK

**Adaptive Gradient Compression Strategy.** Adaptive strategies address the convergence issues of static sparsity or quantization by dynamically adjusting compression based on training dynamics, device heterogeneity, or network conditions. DC2 Abdelmoniem & Canini (2021a) adapts communication frequency to bandwidth but does not co-optimize the compression ratio $\delta$ or consider convergence coupling. DAGC Lu et al. (2023a) optimizes $\delta$ based on client data volume but assumes a fixed local update frequency $\tau$. L-GreCo Markov et al. (2024) solves layer-wise compression allocation as a constrained optimization problem, but ignores local update or network conditions. Shadowheart SGD Tyurin et al. (2024) introduces a time-equilibrium strategy that dynamically selects compression

Table 2: Training time (in seconds) to reach target accuracy (VGG@Flickr) or perplexity (GPT@Wiki) under different $\phi_c$ values, where $\phi(\tau, \delta) = \frac{2^\tau}{\delta^2}$. Results show that WingsFL achieves faster convergence and is robust to $\phi_c$ variations.

| Task | $\phi_c$ | Target | FedAVG | $\gamma$-FedHT | PASGD | WingsFL |
|---|---|---|---|---|---|---|
| VGG@Flickr | $\phi(25, 0.1\%)$ | 55% | 310.69 | 302.82 | 284.22 | **195.94** |
| | | 60% | 396.20 | 356.50 | 339.02 | **246.13** |
| | | 65% | 553.26 | 449.89 | 428.94 | **336.93** |
| | $\phi(25, 1\%)$ | 55% | 354.97 | 229.42 | 274.52 | **192.36** |
| | | 60% | 453.00 | 281.89 | 330.64 | **244.30** |
| | | 65% | 632.77 | 362.98 | 418.98 | **327.89** |
| | $\phi(30, 0.1\%)$ | 55% | 323.01 | 312.79 | 306.07 | **206.27** |
| | | 60% | 403.50 | 370.00 | 365.38 | **259.17** |
| | | 65% | 548.47 | 455.17 | 459.11 | **353.92** |
| | $\phi(30, 1\%)$ | 55% | 278.59 | 236.62 | 299.31 | **195.94** |
| | | 60% | 358.92 | 289.46 | 355.79 | **249.05** |
| | | 65% | 495.60 | 367.37 | 448.49 | **345.57** |
| GPT@Wiki | $\phi(25, 0.1\%)$ | 23.5 | 82.89 | 123.11 | 100.37 | **67.35** |
| | | 23.0 | 113.77 | 201.44 | 156.23 | **89.79** |
| | | 22.5 | 217.79 | 301.94 | 230.65 | **157.88** |
| | $\phi(25, 1\%)$ | 23.5 | 82.97 | 88.43 | 100.51 | **62.22** |
| | | 23.0 | 113.80 | 123.80 | 146.91 | **92.87** |
| | | 22.5 | 217.66 | 185.63 | 239.56 | **151.70** |
| | $\phi(30, 0.1\%)$ | 23.5 | 74.18 | 127.34 | 118.30 | **63.86** |
| | | 23.0 | 125.39 | 203.69 | 174.29 | **102.17** |
| | | 22.5 | 190.36 | 305.48 | 263.88 | **153.76** |
| | $\phi(30, 1\%)$ | 23.5 | 106.32 | 93.07 | 118.95 | **61.89** |
| | | 23.0 | 177.18 | 124.76 | 164.17 | **103.16** |
| | | 22.5 | 253.29 | 187.10 | 253.66 | **151.67** |

Table 3: Training time (in seconds) to reach target accuracy (VGG@Flickr) or perplexity (GPT@Wiki) under different device heterogeneity levels $q\%$, where $t_{\text{compute}}^i \in [t, (1 + q\%)t]$ and $t_{\text{compute}}^i$ is task-specific. Results demonstrate WingsFL's robustness to device heterogeneity.

| Task | Heterogeneity | Target | FedAVG | $\gamma$-FedHT | PASGD | WingsFL |
|---|---|---|---|---|---|---|
| VGG@Flickr | 20% | 55% | 206.48 | 144.15 | 183.22 | **125.02** |
| | | 60% | 266.02 | 173.98 | 218.56 | **157.69** |
| | | 65% | 362.27 | 231.91 | 275.04 | **212.38** |
| | 50% | 55% | 233.40 | 179.44 | 228.35 | **149.36** |
| | | 60% | 300.71 | 219.49 | 269.58 | **189.73** |
| | | 65% | 415.20 | 278.55 | 341.29 | **263.50** |
| | 100% | 55% | 278.59 | 236.62 | 299.31 | **195.94** |
| | | 60% | 358.92 | 289.46 | 355.79 | **249.05** |
| | | 65% | 495.60 | 367.37 | 448.49 | **345.57** |
| | 200% | 55% | 368.97 | 350.97 | 441.22 | **289.10** |
| | | 60% | 475.34 | 429.40 | 528.20 | **367.70** |
| | | 65% | 656.39 | 545.00 | 662.88 | **509.71** |
| GPT@Wiki | 20% | 23.5 | 81.97 | 57.41 | 72.85 | **42.30** |
| | | 23.0 | 129.60 | 76.55 | 100.26 | **63.91** |
| | | 22.5 | 210.45 | 114.78 | 155.01 | **93.77** |
| | 50% | 23.5 | 91.11 | 71.12 | 90.41 | **47.39** |
| | | 23.0 | 151.83 | 94.83 | 124.38 | **78.97** |
| | | 22.5 | 222.05 | 142.19 | 183.75 | **114.82** |
| | 100% | 23.5 | 106.32 | 93.07 | 118.95 | **61.89** |
| | | 23.0 | 177.18 | 124.76 | 164.17 | **103.16** |
| | | 22.5 | 253.29 | 187.10 | 253.66 | **151.67** |
| | 200% | 23.5 | 136.73 | 139.57 | 177.03 | **93.04** |
| | | 23.0 | 227.87 | 186.09 | 243.76 | **155.05** |
| | | 22.5 | 315.76 | 279.09 | 410.53 | **228.51** |

parameters based on heterogeneous computation and communication delays, achieving optimal time complexity. Overall, most methods optimize either $\delta$ or $\tau$ independently, ignoring their joint impact on convergence.

**Federated Learning with Heterogeneous Devices.** To address device heterogeneity, HeteroFL Diao et al. (2021) enables clients to train subnetworks of varying sizes. FIARSE Wu et al. (2024) further extracts submodels based on parameter importance, allowing adaptive training across devices with weighted server aggregation. From a system perspective, asynchronous or hybrid aggregation schemes mitigate straggler effects. FedBuff Nguyen et al. (2022) combines synchronous and asynchronous buffering to boost efficiency. These approaches, however, typically decouple gradient compression and communication frequency. WingsFL addresses this gap by jointly optimizing both, enhancing end-to-end performance with minimal overhead.

**Adaptive Infrequent Communication.** Adaptive local update schemes reduce communication cost based on local-global model similarity. The work Shenaj et al. (2025) proposes continuing local updates when local and global representations are aligned, terminating early when divergence is high. AQUILA Zhao et al. (2022) adaptively adjusts quantization accuracy and communication frequency, combining lazy aggregation with theoretical guarantees to reduce transmission overhead while maintaining convergence. These strategies show that communication schedules can be effectively tailored based on model dynamics and feedback signals.

## 7 CONCLUSION

FL usually uses infrequent communication and gradient compression to alleviate the communication bottlenecks, which severally prevents the training process facing low bandwidth. We demonstrate that the effect of these two strategies on the model convergence are not orthogonal but intrinsically coupled. We proposed WingsFL, a novel theoretical framework that reveals the term $\frac{2^\tau}{\delta^2}$ as a key factor governing the convergence of FedAVG with gradient compression. Leveraging this insight, we propose WingsFLthat jointly optimizes the compression ratio $\delta$ and the number of local steps $\tau$ to minimize end-to-end training time under device heterogeneity and dynamic bandwidth. Our design achieves up to $2.24\times$ and $2.18\times$ speed-ups over FedAVG and SOTA adaptive algorithms, showing that our approach significantly outperforms static and adaptive baselines by efficiently co-optimizing these hyperparameters.

## 8 ETHICS STATEMENT

This research adheres to the ICLR Code of Ethics . No human subjects, personal data, or sensitive demographic attributes were involved in this study. All datasets used are publicly available and do not contain personally identifiable information. The methodology proposed is not intended to produce harmful outputs or be deployed in high-stakes decision-making contexts without further safety evaluation. There are no known risks of discrimination, bias, or unfairness associated with our approach. The authors declare no conflicts of interest or funding-related bias. All experimental procedures comply with standard ethical research practices.

## 9 REPRODUCIBILITY STATEMENT

To support reproducibility, we provide detailed descriptions of model architectures, training protocols, and hyperparameter settings in the main paper and Appendix. All theoretical results are accompanied by complete proofs (in Appendix D). The source code used to run all experiments will be published if accepted. Public datasets are used and provided in Section 5. These resources collectively enable researchers to reproduce the main results presented in this work.

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

CONTENTS

## A   THE USE OF LARGE LANGUAGE MODELS(LLMS)

In this work, LLMs are used to polish the lareadability and find the related works. LLMs are not used for the generation of ideas, experimental design, data analysis, or any other part of the research process.

## B   NOTATION LIST

Table 4: Notation list.

| Notation | Description |
|---|---|
| $n$ | number of workers |
| $T_{\text{end}}$ | total number of global iterations |
| $f_i(\cdot)$ | the local loss function of worker $i$ |
| $p_i$ | the training weight of worker $i$, typically proportional to its local dataset volume |
| $f(\cdot)$ | the global loss function, *i.e.*, $f(\mathbf{x}) = \sum_{i=1}^{n} p_i f_i(\mathbf{x})$ |
| $\delta$ | compression ratio ($0 < \delta \le 1$) |
| $\tau$ | communication infrequency |
| $\mathbf{C}_\delta(\cdot)$ | the sparsification compressor with compression ratio $\delta$ |
| $\mathbf{x}_{T,j}^i$ | the local model parameter in the $T$-th global iteration, $j$-th local iteration of the $i$-th worker |
| $\mathbf{x}_T$ | the global model parameter in the $T$-th global iteration |
| $\mathbf{g}_{T,j}^i$ | the stochastic gradient of $\mathbf{x}_{T,j}^i$ |
| $\Delta_T^i$ | the sum of the gradients within $\tau$ local iterations in the $T$-th global iteration of worker $i$ |
| $\gamma_T$ | the stepsize at global iteration $T$ |
| $\mathbf{e}_T^i$ | the local error term of the $i$-th worker at the $T$-th global iteration |
| $L$ | L-smoothness |
| $\mu$ | $\mu$-strongly convexity if $f_i$ is strongly convex |
| $\zeta$ | the global data heterogenority |
| $\xi^i$ | the stochastic gradient noise of worker $i$ |
| $\sigma^2$ | the upper variance bound of the stochastic gradient noise |
| $a^i$ | the coeffcient parameter of the communication cost (s) |
| $b_T^i$ | the bandwidth of the link between worker $i$ and server at the global iteration $T$ (bits/s) |
| $S_g$ | the size of the gradient (bits) |
| $t_{\text{latency}}^i$ | the network end-to-end latency of worker $i$ (s) |
| $t_{\text{compute}}^i$ | the time for a single computation pass of worker $i$ (s) |
| $T_{avg}$ | average end-to-end training time per iteration (s) |

## C  FEDAVG-GC

---

**Algorithm 3:** FedAVG-GC

---

**Input:** number of clinets $n$, traing weight $p_i$, step-size $\gamma$, initial parameters $\mathbf{x}_0$, initial local error
$\quad\quad \mathbf{e}_0^i = \mathbf{0}_d$, the communication frequency $\tau$, the compressor $\mathbf{C}_\delta(\cdot)$ with the compression
$\quad\quad$ ratio $\delta$

**Output:** $\mathbf{x}_{T_{\text{end}}}$

**for** $T \in [T_{end}]$ **do**

$\quad$ /* Worker side $\quad\quad\quad\quad\quad\quad\quad\quad\quad\quad\quad\quad\quad\quad\quad\quad\quad\quad\quad\quad\quad\quad\quad\quad$ */

$\quad$ **for** $i \in [n]$ **do**

$\quad\quad$ $\mathbf{x}_{T,0}^i = \mathbf{x}_T$;

$\quad\quad$ **for** $j \in [\tau]$ **do**

$\quad\quad\quad$ $\mathbf{x}_{T,j+1}^i = \mathbf{x}_{T,j}^i - \gamma \mathbf{g}_{T,j}^i$;

$\quad\quad$ **end**

$\quad\quad$ $\Delta_T^i = \sum_{j \in [\tau]} \mathbf{g}_{T,j}^i$;

$\quad\quad$ $\hat{\Delta}_T^i = \mathbf{C}_\delta(\Delta_T^i + \mathbf{e}_T^i)$;

$\quad\quad$ $\mathbf{e}_{T+1}^i = \mathbf{e}_T^i + \Delta_T^i - \hat{\Delta}_T^i$;

$\quad\quad$ Upload $\hat{\Delta}_T^i$ to the server;

$\quad$ **end**

$\quad$ /* Server side $\quad\quad\quad\quad\quad\quad\quad\quad\quad\quad\quad\quad\quad\quad\quad\quad\quad\quad\quad\quad\quad\quad\quad\quad\quad$ */

$\quad$ $\mathbf{x}_{T+1} = \mathbf{x}_T - \gamma \sum_{i \in [n]} p_i \hat{\Delta}_T^i$;

$\quad$ Broadcast $\mathbf{x}_{T+1}$;

**end**

**Return** $\mathbf{x}_{T_{\text{end}}}$;

---

## D  DETAILED PROOF

### D.1  TECHNICAL RESULTS

We list lemmas derived from other works here to help us complete the whole proof. Detailed proof of these lemmas can be found from the reference and we do not write here.

**Lemma 1.** *If $c_1, c_2 \in \mathbb{R}^d$ then the Jensen's inequality is: For all $\rho > 0$, we have*

$$\|c_1 + c_2\|^2 \le (1 + \rho)\|c_1\|^2 + (1 + \rho^{-1})\|c_2\|^2. \tag{11}$$

*This can be written as:*

$$2\langle c_1, c_2 \rangle \le \rho\|c_1\|^2 + \rho^{-1}\|c_2\|^2. \tag{12}$$

**Lemma 2** (The nature of Top-$k$). *By definition, the Top-$k$ compressor $\mathbf{C}_\delta$ is a mapping that has the property $\mathbb{R}^d \to \mathbb{R}^d$:*

$$\mathbb{E}_{\mathbf{C}_\delta}\|\mathbf{C}_\delta(\mathbf{x}) - \mathbf{x}\|^2 \le (1 - \delta)\|\mathbf{x}\|^2. \tag{13}$$

**Lemma 3** (Lemma 27 of the work Stich (2020)). *Let $(r_t)_{t \ge 0}$ and $(s_t)_{t \ge 0}$ be sequences of positive numbers satisfying*

$$r_{t+1} \le r_t - B\gamma s_t + C\gamma^2 + D\gamma^3,$$

*for some positive constants $B > 0$, $C, D \ge 0$ and step-sizes $0 < \gamma \le \frac{1}{E}$, for $E \ge 0$. Then there exists a constant stepsize $\gamma \le \frac{1}{E}$ such that*

$$\frac{B}{T+1} \sum_{t=0}^{T} s_t \le \frac{Er_0}{T+1} + 2D^{1/3}\left(\frac{r_0}{T+1}\right)^{2/3} + 2\left(\frac{Cr_0}{T+1}\right)^{1/2}. \tag{14}$$

**Remark 3.** *To ensure that the right hand side in Eq. 14 is less than $\epsilon > 0$,*

$$T = \mathcal{O}\left(\frac{C}{\epsilon^2} + \frac{\sqrt{D}}{\epsilon^{3/2}} + \frac{E}{\epsilon}\right) \cdot r_0$$

steps are sufficient.

**Lemma 4** (Lemma 25 of the work Stich (2020)). *Let $(r_t)_{t\geq 0}$ and $(s_t)_{t\geq 0}$ be sequences of positive numbers satisfying*

$$r_{t+1} \leq (1 - \min\{\gamma A, F\})r_t - B\gamma s_t + C\gamma^2 + D\gamma^3,$$

*for some positive constants $A, B > 0$, $C, D \geq 0$, and for constant step-sizes $0 < \gamma \leq \frac{1}{E}$, for $E \geq 0$, and for parameter $0 < F \leq 1$. Then there exists a constant step-size $\gamma \leq \frac{1}{E}$ such that*

$$\frac{B}{W_T}\sum_{t=0}^{T} w_t s_t + \min\left\{A, \frac{F}{\gamma}\right\} r_{T+1} \leq r_0\left(E + \frac{A}{F}\right)\exp\left[-\min\left\{\frac{A}{E}, F\right\}(T+1)\right]$$

$$+ \frac{2C\ln\tau}{A(T+1)} + \frac{D\ln^2\tau}{A^2(T+1)^2}$$

*for $w_t := (1 - \min\{\gamma A, F\})^{-(t+1)}$, $W_T := \sum_{t=0}^{T} w_t$ and*

$$\tau = \max\left\{\exp[1], \min\left\{\frac{A^2 r_0(T+1)^2}{C}, \frac{A^3 r_0(T+1)^3}{D}\right\}\right\}.$$

**Remark 4.** *Lemma 4 establishes a bound of the order*

$$\tilde{\mathcal{O}}\left(r_0\left(E + \frac{A}{F}\right)\exp\left[-\min\left\{\frac{A}{E}, F\right\}T\right] + \frac{C}{AT} + \frac{D}{A^2 T^2}\right),$$

that decreases with $T$. To ensure that this expression is less than $\epsilon$,

$$T = \tilde{\mathcal{O}}\left(\frac{C}{A\epsilon} + \frac{\sqrt{D}}{A\sqrt{\epsilon}} + \frac{1}{F}\log\frac{1}{\epsilon} + \frac{E}{A}\log\frac{1}{\epsilon}\right) = \tilde{\mathcal{O}}\left(\frac{C}{A\epsilon} + \frac{\sqrt{D}}{A\sqrt{\epsilon}} + \frac{1}{F} + \frac{E}{A}\right)$$

steps are sufficient.

### D.2 Key Lemmas to Prove Theorem 1

In this section, we prove Theorem 1. We follow the analysis of Section 3.2 and define two virtual sequences:

$$\tilde{\mathbf{x}}_0 = \hat{\mathbf{x}}_{0,0} = \mathbf{x}_0, \quad \tilde{\mathbf{x}}_{T+1} = \tilde{\mathbf{x}}_T - \gamma\sum_{i\in[n]} p_i\Delta_T^i, \quad \hat{\mathbf{x}}_{T,j+1} = \hat{\mathbf{x}}_{T,j} - \gamma\sum_{i\in[n]} p_i\mathbf{g}_{T,j}^i. \tag{15}$$

In addition, we use the notation $\hat{F}_T = \sum_{j\in[\tau]} \hat{F}_{T,j}$, $\hat{F}_{T,j} = \mathbb{E}f(\hat{\mathbf{x}}_{T,j}) - f^*$, $G_T = \sum_{j\in[\tau]} \|\nabla f(\mathbf{x}_{T,j})\|^2$, $\tilde{B}_{T,j} = \|\hat{\mathbf{x}}_{T,j} - \tilde{\mathbf{x}}_{T,j}\|^2$, $B_{T,j} = \|\mathbf{x}_{T,j} - \tilde{\mathbf{x}}_{T,j}\|^2$, $\tilde{B}_T = \sum_{j\in[\tau]} \tilde{B}_{T,j}$, $B_T = \sum_{j\in[\tau]} B_{T,j}$, $E_T = n\sum_{i\in[n]} p_i^2\|\mathbf{e}_T^i\|^2$, $\hat{\mathbf{x}}_{T,j} = \mathbb{E}\|\hat{\mathbf{x}}_{T,j} - \mathbf{x}^*\|^2$ and $\hat{\mathbf{x}}_T = \sum_{j\in[\tau]} \hat{\mathbf{x}}_{T,j}$. $\mathbf{x}^x$ is the optimal model parameter, that is $f^* = \min_{\mathbf{x}\in\mathbb{R}^d} f(\mathbf{x}) = f(\mathbf{x}^*)$.

**Lemma 5.** *Let $f$ be $L$-smooth. If $\gamma \leq \frac{1}{2L}$, then it holds for the iterates of FedAVG-GC:*

$$\hat{F}_{T+1} \leq \hat{F}_T - \frac{\gamma}{4}G_T + \frac{\gamma^2 L\tau\sum_{i\in[n]} p_i^2\sigma^2}{2} + \gamma L^2(\tilde{B}_T + \gamma^2\tau E_T). \tag{16}$$

*Proof.*

$$\hat{F}_{T+1,j} \leq \hat{F}_{T,j} - \langle\gamma\sum_i p_i\mathbf{g}_{T,j}^i, \nabla f(\hat{\mathbf{x}}_{T,j})\rangle + \frac{\gamma^2 L}{2}\|\sum_i p_i\mathbf{g}_{T,j}^i\|^2$$

$$\leq \hat{F}_{T,j} - \langle\gamma\nabla f(\mathbf{x}_{T,j}), \nabla f(\hat{\mathbf{x}}_{T,j})\rangle + \frac{\gamma^2 L}{2}(\|\nabla f(\mathbf{x}_{T,j})\|^2 + \sum_i p_i^2\sigma^2)$$

$$\leq \hat{F}_{T,j} - \frac{\gamma}{2}\|\nabla f(\mathbf{x}_{T,j})\|^2 + \frac{\gamma}{2}(\|\nabla f(\mathbf{x}_{T,j}) - \nabla f(\hat{\mathbf{x}}_{T,j})\|) \tag{17}$$

$$+ \frac{\gamma^2 L}{2}\|\nabla f(\mathbf{x}_{T,j})\|^2 + \frac{L\gamma^2\sum_i p_i^2\sigma^2}{2}$$

$$\leq \hat{F}_{T,j} - \frac{\gamma}{2}(1 - L\gamma)\|\nabla f(\mathbf{x}_{T,j})\|^2 + \gamma L^2(\|\hat{\mathbf{x}}_{T,j} - \tilde{\mathbf{x}}_{T,j}\|^2 + \|\tilde{\mathbf{x}}_{T,j} - \mathbf{x}_{T,j}\|^2)$$

$$+ \frac{L\gamma^2\sum_i p_i^2\sigma^2}{2},$$

where the first inequality is due to $L$-smoothness (Assumption 1), and the second inequality is due to the assumption on the gradient noise (Assumption 2). The third is due to Eq. 12 and the fourth is due to Eq. 12 and $L$-smoothness.

We observe that $B_T = \tau\|\gamma \sum_i p_i \mathbf{e}_T^i\|^2 \leq n\tau\gamma^2 \sum_i p_i^2 \|\mathbf{e}_T^i\|^2 = \gamma^2 \tau E_T$. We sum Eq. 17 from $j = 1$ to $j = \tau$ and have

$$\hat{F}_{T+1} \overset{\gamma < \frac{1}{2L}}{\leq} \hat{F}_T - \frac{\gamma}{4}G_T + \frac{L\gamma^2\tau \sum_i p_i^2 \sigma^2}{2} + \gamma L^2(\tilde{B}_T + \gamma^2\tau E_T).$$

**Lemma 6.** *Let $B_t$ be defined in NVS-FL, we have*

$$\tilde{B}_T \leq \tau^2\gamma^2 G_T + \frac{\tau^2\gamma^2 \sum_i p_i^2 \sigma^2}{2}. \tag{18}$$

*Proof.* Similarly to the previous work Stich & Karimireddy (2020b), we have

$$\begin{aligned}
\tilde{B}_{T,j} &= \|(\hat{\mathbf{x}}_{T,j} - \hat{\mathbf{x}}_{T,0}) - (\tilde{\mathbf{x}}_{T,j} - \hat{\mathbf{x}}_{T,0})\|^2 \\
&\leq \|\tilde{\mathbf{x}}_{T,j} - \hat{\mathbf{x}}_{T,0}\|^2 \\
&= \|\gamma \sum_{j' \in [j]} \sum_i p_i \mathbf{g}_{T,j'}^i\|^2 \\
&\leq \gamma^2 j \sum_{j'} \|\nabla f(\mathbf{x}_{T,j})\|^2 + j \sum_i p_i^2 \sigma^2 \gamma^2.
\end{aligned} \tag{19}$$

The second line is due to $\mathbb{E}\|x - \mathbb{E}x\|^2 \leq \mathbb{E}\|x\|^2$. In this way, we have

$$\begin{aligned}
\tilde{B}_T &= \sum_{j \in [\tau]} \tilde{B}_{T,j} \leq \sum_{j \in [\tau]} \gamma^2 j \sum_{j'} \|\nabla f(\mathbf{x}_{T,j})\|^2 + \sum_i p_i^2 \sigma^2 \gamma^2 \sum_{j \in [\tau]} j \\
&\leq \tau^2\gamma^2 \sum_{j \in [\tau]} \|\nabla f(\mathbf{x}_{T,j})\|^2 + \frac{\tau^2\gamma^2 \sum_i p_i^2 \sigma^2}{2}.
\end{aligned} \tag{20}$$

**Lemma 7.** *It holds for the update of error terms in FedAVG-GC*

$$E_{T+1} \leq (1 - \frac{\delta}{2\tau})E_T + \frac{2\tau Z^2}{\delta} \cdot G_T + \tau(\frac{2\zeta^2}{\delta} + n\sum_i p_i^2 \sigma^2). \tag{21}$$

*Proof.*

$$\begin{aligned}
\|\mathbf{e}_{T+1}^i\|^2 &\leq (1 - \delta)\|\mathbf{e}_T^i + \Delta_T^i\|^2 \\
&\leq (1 - \delta)\|\mathbf{e}_T^i + \sum_j \nabla f_i(\mathbf{x}_{T,j})\|^2 + (1 - \delta)\tau\sigma^2 \\
&\leq (1 - \delta)(1 + \rho)\|\mathbf{e}_T^i\|^2 + (1 - \delta)(1 + \rho^{-1})\|\sum_{j \in [\tau]} \nabla f_i(\mathbf{x}_{T,j})\|^2 + (1 - \delta)\tau\sigma^2 \\
&\leq (1 - \frac{\delta}{2\tau})\|\mathbf{e}_T^i\|^2 + \frac{2\tau}{\delta}\sum_{j \in [\tau]} \|\nabla f_i(\mathbf{x}_{T,j})\|^2 + \tau\sigma^2.
\end{aligned} \tag{22}$$

The first line is due to Eq. 13, the second line is due to Assumption 2, the forth line is due to Eq. 1 and in the forth line, we take $\rho = \frac{2^\tau - 1}{2^\tau} \cdot \frac{\delta}{1 - \delta}$. Then we have

$$E_{T+1} \leq (1 - \frac{\delta}{2\tau})E_T + \frac{2\tau}{\delta}(Z^2\|\nabla f(\mathbf{x}_{T,j})\|^2 + \zeta^2) + n\tau\sum_i p_i^2 \sigma^2, \tag{23}$$

where the inequality is due to Assumption 3.

**Lemma 8.** *Let $f$ be $L$-smooth and $\gamma \leq \{\frac{1}{2\sqrt{2\tau}L}, \frac{\delta}{\sqrt{32 \cdot 2^\tau \tau L Z}}\}$, then it holds*

$$\Xi_{T+1} \leq \Xi_T - \frac{\gamma}{16}G_T + c_1\gamma^2 + c_2\gamma^3, \tag{24}$$

*for $\Xi_T = \hat{F}_T + bE_T$, where $b = \frac{\gamma^3\tau L^2 2^\tau}{\delta}$, $c_1 = \frac{L\tau \sum_i p_i^2 \sigma^2}{2}$, $c_2 = \tau L c_1 + \frac{\tau^2 L^2 2^\tau}{\delta}(\frac{2\zeta^2}{\delta} + n\sum_i p_i^2 \sigma^2)$.*

*Proof.*

$$\Xi_{T+1} = \hat{F}_{T+1} + bE_{T+1}$$

$$\leq \hat{F}_T + bE_T - \frac{\gamma}{8}(1 - \frac{8b}{\gamma} \cdot \frac{2\tau Z^2}{\delta})G_T + c_1\gamma^2 + c_2\gamma^3 \tag{25}$$

$$\leq \Xi_T - \frac{\gamma}{16}G_T + c_1\gamma^2 + c_2\gamma^3,$$

where the second line is due to Lemma 5, 6, 7 and the last inequality is due to $\gamma \leq \frac{\delta}{\sqrt{32 \cdot 2^\tau}\tau LZ}$.

**Theorem 1** (Non-convex convergence rate of FedAVG-GC). *Let $f : \mathbb{R}^d \to \mathbb{R}$ be L-smooth. There exists a stepsize $\gamma \leq \min\{\frac{1}{2\sqrt{2}\tau L}, \frac{1}{\sqrt{32\phi}\tau LZ}\}$, where $\phi = \frac{2^\tau}{\delta^2}$, such that at most*

$$\mathcal{O}(\frac{\sum_{i\in[n]} p_i^2\sigma^2}{\epsilon^2} + \frac{\sqrt{\phi\zeta^2 + (1 + n\phi\delta)\sum_{i\in[n]} p_i^2\sigma^2}}{\epsilon^{3/2}} + \frac{1}{\epsilon} + \frac{Z\sqrt{\phi}}{\epsilon}) \cdot L(f(\mathbf{x}_0) - f^*) \tag{26}$$

*iterations of FedAVG-GC, it holds $\mathbb{E}\|\nabla f(\mathbf{x}_{out})\|^2 \leq \epsilon$, and $\mathbf{x}_{out} = \mathbf{x}_t$ denotes an iterate $\mathbf{x}_t \in \{\mathbf{x}_{0,0}, \dots, \mathbf{x}_{T_{end},0}\}$, where $\tau \cdot T_{end}$ denotes the total number of local iterations, chosen at random uniformly.*

*Proof.* We take Lemma 8 to Lemma 3 and let $r_t = \Xi_T$, $s_t = G_T$, $B = \frac{1}{16}$, $C = c_1$, $D = c_2$, $E = \max\{2\sqrt{2}\tau L, \sqrt{32\phi}\tau LZ\}$. To ensure $\frac{B}{T_{end}}\frac{1}{\tau}\sum_{t\in[T_{end}]} s_t$ is less than $\epsilon > 0$, $T_{end} = \mathcal{O}(\frac{\sum_{i\in[n]} p_i^2\sigma^2}{\epsilon^2} + \frac{\sqrt{\phi\zeta^2 + (1+n\phi\delta)\sum_{i\in[n]} p_i^2\sigma^2}}{\epsilon^{3/2}} + \frac{1}{\epsilon} + \frac{Z\sqrt{\phi}}{\epsilon}) \cdot \frac{L\tau(f(\mathbf{x}_0) - f^*)}{\tau}$ are enough, which completes the proof.

### D.3 Key Lemmas to Prove Theorem 2

Let $F_{T,j} = f(\mathbf{x}_{T,j}) - f^*$ and $F_T = \sum_{j\in[\tau]} F_{T,j}$.

**Lemma 9.** *Let $f$ be L-smooth and $\mu$-convexity, if $\gamma \leq \min\{\frac{1}{4L}, \frac{1}{4\sqrt{3}L\tau}\}$, then it holds for the iterates of FedAVG-GC*

$$\hat{\mathbf{x}}_{T+1} \leq (1 - \frac{\mu\gamma}{2})\hat{\mathbf{x}}_T - \frac{\gamma}{4}F_T + c_1\gamma^2 + c_2\gamma^3 + 6\gamma^3 L\tau E_T, \tag{27}$$

*where $c_1 = \tau \sum p_i^2\sigma^2$ and $c_2 = 3L\tau^2 \sum p_i^2\sigma^2$.*

*Proof.*

$$\hat{\mathbf{x}}_{T+1,j} = \|\hat{\mathbf{x}}_{T,j} - \mathbf{x}^*\|^2 - 2\langle\hat{\mathbf{x}}_{T,j} - \mathbf{x}^*, \gamma\mathbf{g}_{T,j}\rangle + \|\mathbf{g}_{T,j}\|^2$$

$$= \|\hat{\mathbf{x}}_{T,j} - \mathbf{x}^*\|^2 - 2\gamma\langle\nabla f(\mathbf{x}_{T,j}), \mathbf{x}_{T,j} - \mathbf{x}^*\rangle + \|\mathbf{g}_{T,j}\|^2 + 2\gamma\langle\nabla f(\mathbf{x}_{T,j}), \mathbf{x}_{T,j} - \hat{\mathbf{x}}_{T,j}\rangle. \tag{28}$$

Then we analyze the term $2\langle\nabla f(\mathbf{x}_{T,j}), \mathbf{x}_{T,j} - \hat{\mathbf{x}}_{T,j}\rangle$ and $-2\langle\nabla f(\mathbf{x}_{T,j}), \mathbf{x}_{T,j} - \mathbf{x}^*\rangle$:

$$2\langle\nabla f(\mathbf{x}_{T,j}), \mathbf{x}_{T,j} - \hat{\mathbf{x}}_{T,j}\rangle \leq \frac{1}{2L}\|\nabla f(\mathbf{x}_{T,j})\|^2 + 2L\|\mathbf{x}_{T,j} - \hat{\mathbf{x}}_{T,j}\|^2$$

$$\leq f(\mathbf{x}_{T,j}) - f^* + 2L\|\mathbf{x}_{T,j} - \hat{\mathbf{x}}_{T,j}\|^2, \tag{29}$$

where the first line is due to Eq. 12 and the second line is due to $L$-smoothness.

$$-2\langle\nabla f(\mathbf{x}_{T,j}), \mathbf{x}_{T,j} - \mathbf{x}^*\rangle \leq -2(f(\mathbf{x}_{T,j}) - f^*) - \mu\|\mathbf{x}_{T,j} - \mathbf{x}^*\|^2$$

$$\leq -2(f(\mathbf{x}_{T,j}) - f^*) - \frac{\mu}{2}\|\hat{\mathbf{x}}_{T,j} - \mathbf{x}^*\|^2 + \mu\|\hat{\mathbf{x}}_{T,j} - \mathbf{x}_{T,j}\|^2, \tag{30}$$

where the first line is due to Assumption 4 and the second line is due to Eq. 11.

Then we take Eq. 29 and 30 into Eq. 28, then we have

$$\hat{\mathbf{x}}_{T+1,j} \leq \hat{\mathbf{x}}_{T,j} - \gamma(f(\mathbf{x}_{T,j}) - f^*) - \frac{\mu\gamma}{2}\hat{\mathbf{x}}_{T,j} + \gamma(2L + \mu)\|\hat{\mathbf{x}}_{T,j} - \mathbf{x}_{T,j}\|^2$$

$$+ \gamma^2(\|\nabla f(\mathbf{x}_{T,j})\|^2 + \sum p_i^2\sigma^2)$$

$$\leq (1 - \frac{\mu\gamma}{2})\hat{\mathbf{x}}_{T,j} - \gamma(1 - 2\gamma L)F_{T,j} + \gamma(2L + \mu)\|\hat{\mathbf{x}}_{T,j} - \mathbf{x}_{T,j}\|^2 + \gamma^2\sum p_i^2\sigma^2 \tag{31}$$

$$\leq (1 - \frac{\mu\gamma}{2})\hat{\mathbf{x}}_{T,j} - \frac{\gamma}{2}F_{T,j} + 6L\gamma(\tilde{B}_{T,j} + \|\tilde{\mathbf{x}}_{T,j} - \mathbf{x}_{T,j}\|^2) + \gamma^2\sum p_i^2\sigma^2,$$

where the second line is due to $L$-smooth and the last line is due to $\mu \leq L$ (combining Assumption 1 and 4). Then we substitute it and have

$$\hat{\mathbf{x}}_{T+1} \leq (1 - \frac{\mu\gamma}{2})\hat{\mathbf{x}}_T - \frac{\gamma}{2}F_T + 6L\gamma(\tilde{B}_T + B_T) + \gamma^2\tau\sum p_i^2\sigma^2$$
$$\leq (1 - \frac{\mu\gamma}{2})\hat{\mathbf{x}}_T - \frac{\gamma}{2}F_T + 6L\gamma(\tilde{B}_T + \gamma^2\tau E_T) + \gamma^2\tau\sum p_i^2\sigma^2. \tag{32}$$

We take Lemma 6 into Eq. 32 and use $G_T \leq 2LF_T$ based on the $L$-smoothness assumption, then we have

$$\hat{\mathbf{x}}_{T+1} \leq (1 - \frac{\mu\gamma}{2})\hat{\mathbf{x}}_T - \frac{\gamma}{2}(1 - 24\gamma^2L^2\tau^2)F_T + c_1\gamma^2 + c_2\gamma^3 + 6\gamma^3L\tau E_T$$
$$\leq (1 - \frac{\mu\gamma}{2})\hat{\mathbf{x}}_T - \frac{\gamma}{4}F_T + c_1\gamma^2 + c_2\gamma^3 + 6\gamma^3L\tau E_T, \tag{33}$$

where $c_1 = \tau\sum p_i^2\sigma^2$ and $c_2 = 3L\tau^2\sum p_i^2\sigma^2$, completing the proof.

**Lemma 10.** *Let $\Psi_T = \hat{\mathbf{x}}_T + aE_T$, where $a = \frac{12\gamma^3\tau L2^\tau}{\delta}$*

$$\Psi_{T+1} \leq (1-c)\Psi_T - \frac{\gamma}{8}F_T + c_1\gamma^2 + c_2'\gamma^3, \tag{34}$$

*where $c = \min(\frac{\mu\gamma}{2}, \frac{\delta}{2\cdot2^\tau})$, $c_1 = \tau\sum p_i^2\sigma^2$, $c_2' = 3L\tau c_1 + \frac{12L\tau^2 2^\tau}{\delta}(\frac{2\zeta^2}{\delta} + n\sum_i p_i^2\sigma^2)$ with $\gamma \leq \frac{\delta}{\sqrt{384\cdot2^\tau}\tau LZ}$*

*Proof.*

$$\Psi_{T+1} \leq (1 - \frac{\mu\gamma}{2})\hat{\mathbf{x}}_T + a(1 - \frac{\delta}{2^\tau} + \frac{6\gamma^3L\tau}{a})E_T + (\frac{4\tau LZ^2a}{\delta} - \frac{\gamma}{4})F_T$$
$$+ c_1\gamma^2 + c_2\gamma^3 + a\tau(\frac{2\zeta^2}{\delta} + n\sum_i p_i^2\sigma^2) \tag{35}$$
$$\leq (1-c)\hat{\mathbf{x}}_T - \frac{\gamma}{8}F_T + c_1\gamma^2 + c_2'\gamma^3,$$

where the first inequality is due to Lemma 7 and $G_T \leq 2LF_T$, and the second inequality is due to $c = \min(\frac{\mu\gamma}{2}, \frac{\delta}{2\cdot2^\tau})$ and $\gamma \leq \frac{\delta}{\sqrt{384\cdot2^\tau}\tau LZ}$. In this way, we complete the proof.

**Theorem 2** (Convex convergence rate of FedAVG-GC, *i.e.*, $\mu = 0$ ). *Let $f : \mathbb{R}^d \to \mathbb{R}$ be $L$-smooth and $\mu$-convex. Then there exists a stepsize $\gamma \leq \min\{\frac{1}{4L}, \frac{1}{4\sqrt{3}L\tau}, \frac{1}{\sqrt{384\cdot\phi}\tau LZ}\}$, where $\phi = \frac{2^\tau}{\delta^2}$, such that at most*

$$\mathcal{O}(\frac{\sum_{i\in[n]}p_i^2\sigma^2}{\epsilon^2} + \frac{\sqrt{L\phi\zeta^2 + L(1 + n\phi\delta)\sum_{i\in[n]}p_i^2\sigma^2}}{\epsilon^{3/2}} + \frac{L}{\epsilon} + \frac{ZL\sqrt{\phi}}{\epsilon}) \cdot \|\mathbf{x}_0 - \mathbf{x}_*\|^2 \tag{36}$$

*local iterations of FedAVG-GC, it holds $\mathbb{E}f(\mathbf{x}_{out}) - f^* \leq \epsilon$, and $\mathbf{x}_{out} = \mathbf{x}_t$ denotes an iterate $\mathbf{x}_t \in \{\mathbf{x}_0, \ldots, \mathbf{x}_{(\tau\cdot T_{end}-1)}\}$, where $\tau \cdot T_{end}$ denotes the total number of local iterations, chosen at random uniformly.*

*Proof* When $\mu = 0$, $c = \min(\frac{\mu\gamma}{2}, \frac{\delta}{2\cdot2^\tau}) = 0$, then Lemma 8 can be written into

$$\Psi_{T+1} \leq \Psi_T - \frac{\gamma}{8}F_T + c_1\gamma^2 + c_2'\gamma^3.$$

In this way, the proof is same with that of Theorem 1.

**Theorem 3** ($\mu$-strongly convex convergence rate of FedAVG-GC, *i.e.*, $\mu > 0$). *Let $f : \mathbb{R}^d \to \mathbb{R}$ be $L$-smooth and $\mu$-convex. Then there exists a stepsize $\gamma \leq \min\{\frac{1}{4L}, \frac{1}{4\sqrt{3}L\tau}, \frac{1}{\sqrt{384\cdot\phi}\tau LZ}\}$, where $\phi = \frac{2^\tau}{\delta^2}$, such that at most*

$$\mathcal{O}(\frac{\sum_{i\in[n]}p_i^2\sigma^2}{\mu\epsilon} + \frac{\sqrt{L\phi\zeta^2 + L(1 + n\phi\delta)\sum_{i\in[n]}p_i^2\sigma^2}}{\mu\epsilon^{1/2}} + \frac{L}{\mu} + \frac{ZL\sqrt{\phi}}{\mu}) \cdot \|\mathbf{x}_0 - \mathbf{x}_*\|^2 \tag{37}$$

*local iterations of FedAVG-GC, it holds $\mathbb{E}f(\mathbf{x}_{out}) - f^* \leq \epsilon$, and $\mathbf{x}_{out} = \mathbf{x}_t$ denotes an iterate $\mathbf{x}_t \in \{\mathbf{x}_0, \ldots, \mathbf{x}_{(\tau\cdot T_{end}-1)}\}$, where $\tau \cdot T_{end}$ denotes the total number of local iterations, selected probabilistically based on $(1 - \min\{\frac{\mu\gamma}{2}, \frac{\delta}{2\cdot2^\tau}\})^{-t}$.*

*Proof.* We take Lemma 10 to Lemma 4 and let $r_t = \Psi_T$, $s_t = F_T$, $A = \frac{\mu}{2}$, $F = \frac{\delta}{2\cdot2^\tau}$, $B = \frac{1}{8}$, $C = c_1$, $D = c_2'$, $E = \max\{4L, 4\sqrt{3}\tau L, \sqrt{384\phi}\tau LZ\}$, then the proof completes.

Table 5: Configs in Fig. 1.

| Model@Dataset | $t^i_{\text{compute}}$ (ms) | $t^i_{\text{latency}}$ (ms) | $a^i$ (ms) | $b^i_T$ (MB/s) | $\phi_c$ |
|---|---|---|---|---|---|
| CNN@CIFAR-10 | $12.8 \sim 25.6$ | $0 \sim 20$ | 7.85 | $1 \sim 10$ | $\phi(30,\ 1\%)$ |
| VGG11@Flickr | $13.2 \sim 26.4$ | $0 \sim 20$ | 15.1 | $50 \sim 500$ | $\phi(30,\ 1\%)$ |
| ViT@ImageNet | $282 \sim 584$ | $0 \sim 20$ | 164 | $50 \sim 500$ | $\phi(30,\ 1\%)$ |
| GPT2@Wikitext2 | $132 \sim 264$ | $0 \sim 20$ | 95.0 | $50 \sim 500$ | $\phi(30,\ 1\%)$ |

### D.4 PROOF OF THEOREM 4

*Proof.* We first prove that each individual function $Q^i_T(\tau)$ is strictly convex. The second derivative of $Q^i_T(\tau)$ is given by:

$$(Q^i_T(\tau))'' = \frac{c^i_{3,T} \cdot 2^{\tau/2} \left( (\ln 2 \cdot \tau/2 - 1)^2 + 1 \right) + 2c^i_2}{\tau^3}$$

Since $c^i_{3,T} > 0$, $c^i_2 > 0$, and $\tau \geq 1$, all terms in the numerator are positive. Therefore, $Q''_i(\tau) > 0$ for all $\tau \geq 1$, proving that each $Q^i_T(\tau)$ is strictly convex. Based on the convexity of $Q^i_T(\tau)$, for any $\tau_1, \tau_2 \in [R_\tau]$ and $\lambda \in [0,1]$, and for each $i \in [n]$, we have:

$$Q^i_T(\lambda\tau_1 + (1-\lambda)\tau_2) \leq \lambda Q^i_T(\tau_1) + (1-\lambda)Q^i_T(\tau_2).$$

Furthermore, by the definition of the maximum function, we have: $Q^i_T(\tau_1) \leq Q(\tau_1) \quad \text{and} \quad Q^i_T(\tau_2) \leq Q(\tau_2)$ and have

$$\lambda Q^i_T(\tau_1) + (1-\lambda)Q^i_T(\tau_2) \leq \lambda Q(\tau_1) + (1-\lambda)Q(\tau_2)$$

Combining these results, we obtain for each $i$:

$$Q^i_T(\lambda\tau_1 + (1-\lambda)\tau_2) \leq \lambda Q(\tau_1) + (1-\lambda)Q(\tau_2)$$

Taking the maximum over $i$ on the left-hand side gives:

$$Q(\lambda\tau_1 + (1-\lambda)\tau_2) = \max_{i\in[n]} Q^i_T(\lambda\tau_1 + (1-\lambda)\tau_2) \leq \lambda Q(\tau_1) + (1-\lambda)Q(\tau_2)$$

This proves that $Q(\tau)$ is convex. Since $Q(\tau)$ is convex, any local minimum of $Q(\tau)$ is necessarily a global minimum. This completes the proof. $\square$

## E ADDENDUM TO EVALUATION EXPERIMENTS

### E.1 EXPERIMENTAL ENVIRONMENT

All experiments are performed on a server running Ubuntu 24.04 LTS, equipped with an Intel Xeon Gold 6230 processor and 8 Nvidia RTX 3090 GPUs, each featuring 24GB of VRAM. The software environment is built upon Python 3.10.16, with all dependencies aligned accordingly. We adopt PyTorch 2.5.1 as the core deep learning framework, utilizing CUDA 12.4 for GPU acceleration. Communication across devices is facilitated using the Gloo backend.

### E.2 DETAILED EXPERIMENTAL HYPER-PARAMETERS

The detailed configuration of our experiments in Fig. 1 is presented in Table 5. The computation time $t^i_{\text{compute}}$ for each node is uniformly sampled from the specified range and remains fixed throughout the training process. Similarly, the latency $t^i_{\text{latency}}$ is randomly determined and static. Both parameters are static and do not change during training. The bandwidth $b^i_T$ fluctuates within the given range to simulate a dynamic network environment, where bandwidth varies randomly within this range throughout the training process.

For CNN-based tasks, the model size is significantly smaller (approximately $1/50$ to $1/500$ of the size of other models). To ensure that bandwidth has a noticeable impact on training performance for

such small models, we scale down the bandwidth settings for CNN tasks by a factor of $1/50$ compared to other models.

All subsequent experiments follow this configuration unless otherwise specified. For instance, in the device heterogeneity experiments, we vary the range of $t^i_{\text{compute}}$ while keeping other parameters unchanged. Similarly, for the sensitivity analysis of $\phi_c$, we modify only the $\phi_c$ parameter while maintaining the other settings.

### E.3 DETAILED CURVES OF SECTION 5.4

The detail of Table 2 in the main text of the paper corresponds to Fig. 2 in the Appendix.

### E.4 DETAILED CURVES OF SECTION 5.5

The detail of Table 3 in the main text of the paper corresponds to Fig. 3 in the Appendix.

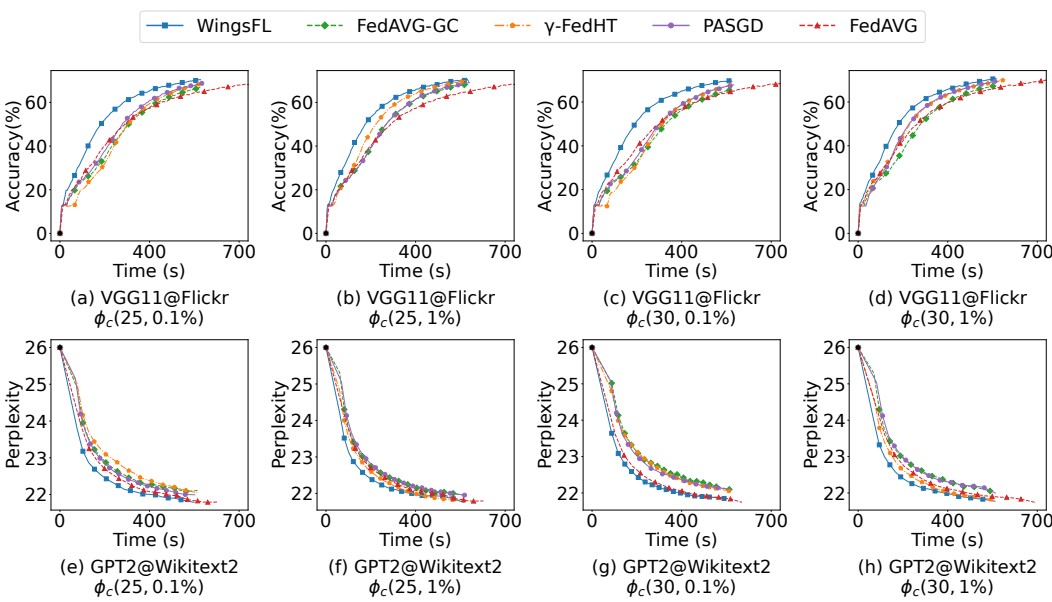

Figure 2: Training curves (Top: VGG11@Flickr; Bottom: GPT2@Wikitext2) across different $\phi_c$ (from left to right).

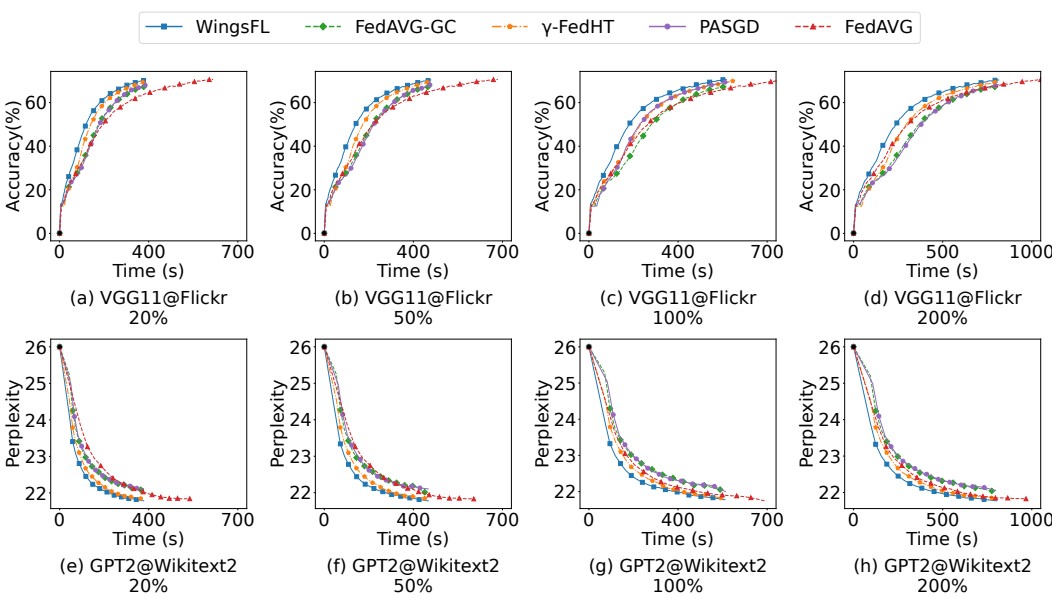

Figure 3: Training curves (Top: VGG11@Flickr; Bottom: GPT2@Wikitext2) across different heterogeneity levels (from left to right).

