# OpenReview forum: "WingsFL: Speed-up Federated Learning via Co-optimization of Communication Frequency and Gradient Compression Ratio"
_ICLR.cc/2026/Conference — ICLR 2026 Conference Withdrawn Submission_

### Official Review · Reviewer_rqfA · 2025-10-20

**Soundness:** 3
**Presentation:** 3
**Contribution:** 3
**Rating:** 4
**Confidence:** 4

**Summary:**

This paper addresses communication bottlenecks in FL by jointly optimizing two key communication strategies: infrequent communication (number of local steps τ) and gradient compression (compression ratio δ) aiming to minimize end-to-end training time. The algorithm formulates this as a Min-Max problem that, using their theoretical equivalence, simplifies to a one-variable optimization solvable via binary search. The core innovation is recognizing the coupling between compression ratio and communication frequency to achieve better end-to-end training performance in FL, particularly under realistic conditions with device heterogeneity and network fluctuations.

**Strengths:**

- The coupling between compression ratio and communication frequency through the 2^τ/δ^2 term is a genuine insight.

- The paper provides a clean systematic way to analyze this coupling.

- The achieved speedups are meaningful for practical FL deployments, especially given the simplicity of the approach.

**Weaknesses:**

- No ablation study isolates the impact of co-optimization vs. single-variable tuning.

- The bandwidth model (uniform random within a range) doesn't capture real network dynamics like packet loss, jitter, or correlated failures common in mobile networks.

- Comparision with related coupling effects like those in asynchronous SGD [Lian et al., 2015] or delayed gradients [Stich & Karimireddy, 2020].

- Binary search overhead and compression costs aren't measured.

- Asynchronous Stochastic Gradient Descent with Delay Compensation, 2015.

- The Error-Feedback Framework: Better Rates for SGD with Delayed Gradients and Compressed Communication, 2020

**Questions:**

Q1: Your entire framework focuses exclusively on gradient sparsification (Top-k). Have you explored whether the theoretical coupling between τ and δ extends to other compression methods (e.g., quantization, sketching, or error feedback with quantization)?

Q2: The paper models total training time as a function of τ, δ, local compute time, and transmission delay. But the real-world cost might also include factors like device dropout, packet loss, or resource contention. Please validate your model against actual system measurements.

Q3: Please provide sensitivity analysis for Hyperparameters \phi.

Q4: Please provide a runtime or overhead analysis for Binary search.

Q5: The algorithm uses “several iterations” to update τ–δ but does not specify how “several” is determined.

Q6: Please provide ablation study to isolate the impact of co-optimization vs. single-variable tuning.

Q7: Please provide the statistical significance and simulation scenarios for more workers.

---

### Official Review · Reviewer_CKYc · 2025-10-29

**Soundness:** 3
**Presentation:** 3
**Contribution:** 2
**Rating:** 4
**Confidence:** 4

**Summary:**

This work explores two key strategies to overcome communication bottlenecks, infrequent communication and gradient compression. In order to optimize the two terms collaboratively, authors first theoretically prove that $\tau$ and $\delta$ are intrinsically coupled and must be co-designed for efficient training. Then WingsFL is proposed to minimize the maximum end-to-end time over all clients. Extensive experiments under device heterogeneity and dynamic network environments across diverse model architectures show the effectiveness of the proposed method.

**Strengths:**

1. The authors propose a novel theoretical framework NVS-FL, to analyze FedAVG-GC. They theoretically establish that the gradient compression and infrequent communication are not orthogonal strategies in FedAVG.

2. The authors mathematically model the end-to-end training time under device heterogeneity and dynamic bandwidth as a one-variable Min-Max problem. And use WingsFL to jointly optimize $\tau$ and $\delta$.

3. Extensive experiments under different settings show the effectiveness of the proposed method.

**Weaknesses:**

1. What’s the challenge in NVS-FL compared to the traditional convergence analysis framework?
2. Considering device heterogeneity, the local steps and compression ratio should be different among all clients, while WingsFL uses one optimal setting for all clients.
3. The experimental evaluation is insufficient. The experiments appear to have been run only once, with no reporting of variance or multiple-seed results, which raises concerns about statistical reliability. No ablation studies are provided to test the effectiveness of the optimal two terms collaboratively, such as stable setting and adaptive setting. And only one gradient compression method is used in the experiment. As stated in Section 6, DAGC, DC2 should be compared in the experiment.
4. Some details need to be improved. For example, the optimization function is a “one-variable” Min-Max problem in line 027 but “dual-variable” in line 072, which is confusing.

**Questions:**

See weaknesses.

---

### Official Review · Reviewer_kqsZ · 2025-10-31

**Soundness:** 2
**Presentation:** 3
**Contribution:** 1
**Rating:** 2
**Confidence:** 4

**Summary:**

This paper proposes a framework that jointly optimizes the communication interval and gradient compression ratio in federated learning. The authors claim that the convergence rate of FedAVG with compression depends on the coupling term $2^{\tau}/ \delta^2$ and derive this through an extension of the Nested Virtual Sequence framework. Building on this, they present WingsFL to accelerate FL training. Experiments on CNN, VGG, ViT, and GPT2 (simulated on a single multi-GPU node) show up to 2.24$\times$ and 2.18$\times$ speed-ups over FedAVG and SOTA adaptive strategies, respectively.

**Strengths:**

1. This paper explores an important practical problem in federated learning: the trade-off between communication frequency and gradient compression. The motivation is clear and based on real-world communication constraints.

2. The paper is well-structured and logically consistent. The flow from theoretical formulation to algorithmic implementation and empirical validation is clear.

3. The experimental section covers a wide range of tasks and models (CNN, VGG, ViT, GPT2), showing consistent performance trends.

**Weaknesses:**

1. This paper contains obvious spelling errors and formula description errors. For example, the strong convexity assumption in Sec. 3.1, namely formula (4), is incorrect. The word “first” in Remark 1 is misspelled, etc. The author should carefully review the entire paper.

2. Citation formatting is inconsistent throughout, and the discussion of prior literature is superficial. The paper references earlier approaches such as DAGC, $\gamma$-FedHT, and Deco-SGD but fails to specify what concrete theoretical or methodological improvement this work achieves. The contribution appears incremental, with no demonstrated advantage in convergence rate, computational
complexity, or general applicability.

3. The conversion from a two-variable minmax problem to a single-variable formulation via fixing $\phi$ lacks theoretical support. The authors do not prove that fixing $\phi$ preserves optimality or convergence equivalence, nor do they provide an analysis of the approximation error induced by this simplification.

4. Theorem 4 assumes that $\tau$ is a continuous variable. In practice, Algorithm 2 operates over a discrete search space. Therefore, the theoretical guarantee of global optimality given by Theorem 4 does not hold for the actual implementation.

5. The experimental section includes only a few baselines (FedAVG, $\gamma$-FedHT, PASGD) and omits many recent relevant methods in adaptive compression or communication-efficient FL, such as [1]-[4]. The paper briefly varies $\phi_c$ but does not explore how $\tau$, $\delta$, or the search frequency $E$ individually affect performance. Without ablation or robustness analysis, it is unclear whether the method’s improvement is consistent across parameter choices.

[1] Zhou H, Lan T, Venkataramani G P, et al. Every parameter matters: Ensuring the convergence of federated learning with dynamic heterogeneous models reduction[J]. Advances in Neural Information Processing Systems, 2023, 36: 25991-26002.

[2] Wang Y, Zhang X, Li M, et al. Theoretical convergence guaranteed resource-adaptive federated learning with mixed heterogeneity[C]//Proceedings of the 29th ACM SIGKDD Conference on Knowledge Discovery and Data Mining. 2023: 2444-2455.

[3] Condat L, Maranjyan A, Richtárik P. LoCoDL: Communication-Efficient Distributed Learning with Local Training and Compression[C]//The Thirteenth International Conference on Learning Representations.

[4] Zhang J, Li N, Dedeoglu M. Federated learning over wireless networks: A band-limited coordinated descent approach[C]//IEEE INFOCOM 2021-IEEE Conference on Computer Communications. IEEE, 2021: 1-10.

**Questions:**

1. The paper cites several related frameworks such as DAGC, $\gamma$-FedHT, and Deco-SGD, yet the conceptual and methodological differences between WingsFL and these prior works are not clearly articulated.
The theoretical contribution of WingsFL therefore appears to be incremental. Could you explicitly identify (a) which theoretical assumption, modeling choice, or optimization step in WingsFL is genuinely new, and (b) what measurable improvement it brings in terms of convergence rate, computational complexity, or applicability under heterogeneous and bandwidth-constrained settings?

2. In addition, the paper should more clearly position its theoretical results relative to other recent studies that analyze communication-efficient or resource-adaptive federated learning, such as Zhou et al. (NeurIPS 2023), Wang et al. (KDD 2023), Condat et al. (ICLR 2025), and Zhang et al. (INFOCOM 2021). Please provide an explicit comparison of your convergence bounds, key assumptions, and theoretical guarantees against these works.

 Zhou H, Lan T, Venkataramani G P, et al. Every parameter matters: Ensuring the convergence of federated learning with dynamic heterogeneous models reduction[J]. Advances in Neural Information Processing Systems, 2023, 36: 25991-26002.

Wang Y, Zhang X, Li M, et al. Theoretical convergence guaranteed resource-adaptive federated learning with mixed heterogeneity[C]//Proceedings of the 29th ACM SIGKDD Conference on Knowledge Discovery and Data Mining. 2023: 2444-2455.

Condat L, Maranjyan A, Richtárik P. LoCoDL: Communication-Efficient Distributed Learning with Local Training and Compression[C]//The Thirteenth International Conference on Learning Representations.

Zhang J, Li N, Dedeoglu M. Federated learning over wireless networks: A band-limited coordinated descent approach[C]//IEEE INFOCOM 2021-IEEE Conference on Computer Communications. IEEE, 2021: 1-10.

3. Can you provide a theoretical argument or at least an approximation bound showing that fixing $\phi$ does not alter the optimal solution? If the equivalence is heuristic, what empirical or analytical evidence supports that it yields comparable or near-optimal results?

---

### Official Review · Reviewer_w21H · 2025-10-31

**Soundness:** 2
**Presentation:** 2
**Contribution:** 2
**Rating:** 2
**Confidence:** 3

**Summary:**

This paper investigates the joint effect of two common communication-efficiency strategies in federated learning (FL): infrequent communication (controlled by the number of local steps, 𝜏) and gradient compression (controlled by the compression ratio, 𝛿). Prior work typically treats these factors as independent and optimizes them separately. The authors challenge this assumption and theoretically establish that the convergence of FedAVG with gradient compression depends on a coupling term $2𝜏/𝛿^2$, showing that 𝜏 and 𝛿 are interdependent.

Building on this insight, they propose WingsFL, a method that co-optimizes 𝜏 and 𝛿 to minimize end-to-end training time under device heterogeneity and dynamic bandwidth conditions. WingsFL converts a two-variable optimization problem into a one-variable Min-Max formulation and solves it efficiently via binary search. Experiments on CNN, VGG, ViT, and GPT-2 models over heterogeneous and dynamic network settings report up to 2.24× speed-up over FedAVG and 2.18× over state-of-the-art adaptive strategies.

**Strengths:**

The paper identifies and analyzes a coupling effect between communication frequency and gradient compression in FL, which is a useful theoretical observation that questions a widely held independence assumption. The reformulation into a single-variable optimization is elegant and computationally lightweight.

**Weaknesses:**

1. The paper claims that $2𝜏/𝛿^2$ governs the convergence rate but does not provide a clear, quantitative interpretation of how communication savings trade off with model accuracy or convergence speed. For example, increasing τ reduces communication but may worsen model divergence; decreasing δ (heavier compression) also degrades gradient fidelity. However, the paper neither visualizes nor explains how WingsFL balances these conflicting forces beyond stating that the key term is “fixed.” Without a clear trade-off analysis, it is hard to evaluate whether WingsFL’s gains come from better communication scheduling or simply from tolerating more approximation error.

2. The algorithmic core, i.e., solving a one-variable minimization based on the derived coupling, is a straightforward adaptation rather than a fundamentally new optimization technique. The insight that τ and δ are not independent is interesting but somewhat incremental, as similar interdependencies have been implicitly discussed in works like PASGD and adaptive compression frameworks.

3. The experimental results mostly report speed-up ratios without showing how convergence rate or final accuracy changes as τ or δ vary. Missing are plots that explicitly demonstrate the communication–accuracy or convergence–communication trade-off. Such analysis would substantiate the claimed “joint optimization.”

4. The heterogeneity setup is synthetic, varying latency and bandwidth in a uniform range. This does not convincingly demonstrate robustness in real heterogeneous or asynchronous FL settings.

**Questions:**

1. Can the authors explicitly show or quantify the trade-off between communication cost and convergence degradation (e.g., accuracy vs. total communication volume) across τ and δ values?

2. Does fixing $2𝜏/𝛿^2$ actually guarantee comparable convergence rates empirically, or is it merely a theoretical construct?

3. What happens if the assumption that τ and δ can be optimized jointly under fixed ϕc is relaxed, does WingsFL still outperform baselines?

4. Could the authors discuss how the binary search algorithm’s overhead compares to the actual training time saved?

5. Would WingsFL generalize to quantization-based or hybrid compression methods, not just Top-k sparsification?

---

### Note · Authors · 2025-11-28

I have read and agree with the venue's withdrawal policy on behalf of myself and my co-authors.